# Substrate promiscuity of inositol 1,4,5-trisphosphate kinase driven by structurally-modified ligands and active site plasticity

María Ángeles Márquez-Moñino [1], Raquel Ortega-García[1], Hayley Whitfield[2,5], Andrew M. Riley [3,5], Lourdes Infantes [1], Shane W. Garrett[4,6], Megan L. Shipton [3], Charles A. Brearley [2], Barry V. L. Potter [3,4] ✉ & Beatriz González [1] ✉

D-*myo*-inositol 1,4,5-trisphosphate (InsP$_3$) is a fundamental second messenger in cellular Ca$^{2+}$ mobilization. InsP$_3$ 3-kinase, a highly specific enzyme binding InsP$_3$ in just one mode, phosphorylates InsP$_3$ specifically at its secondary 3-hydroxyl group to generate a tetrakisphosphate. Using a chemical biology approach with both synthetised and established ligands, combining synthesis, crystallography, computational docking, HPLC and fluorescence polarization binding assays using fluorescently-tagged InsP$_3$, we have surveyed the limits of InsP$_3$ 3-kinase ligand specificity and uncovered surprisingly unforeseen biosynthetic capacity. Structurally-modified ligands exploit active site plasticity generating a helix-tilt. These facilitated uncovering of unexpected substrates phosphorylated at a surrogate extended primary hydroxyl at the inositol pseudo 3-position, applicable even to carbohydrate-based substrates. Crystallization experiments designed to allow reactions to proceed in situ facilitated unequivocal characterization of the atypical tetrakisphosphate products. In summary, we define features of InsP$_3$ 3-kinase plasticity and substrate tolerance that may be more widely exploitable.

The full scope of inositol polyphosphates (InsPs) function impacts on many facets of cell signaling and metabolism including DNA editing, mRNA export, Ca$^{2+}$ signaling, vesicle trafficking, apoptosis and anticancer potential[1,2]. An established paradigm of InsP function is represented by InsP$_3$ (D-*myo*-inositol 1,4,5-trisphosphate), a well-known second messenger involved in Ca$^{2+}$ mobilization. Its binding to InsP$_3$ receptors (IP$_3$R) at the membrane of the endoplasmic reticulum triggers Ca$^{2+}$ release from internal stores. InsP$_3$ is recognized by two enzymes that process this messenger, so terminating Ca$^{2+}$ signals, InsP$_3$ 3-kinase (IP3K) and InsP$_3$ 5-phosphatase (INPP5A). In particular, IP3K is a central protein in mammalian InsP metabolism[3,4]. It phosphorylates

InsP$_3$ yielding D-*myo*-inositol 1,3,4,5-tetrakisphosphate (InsP$_4$) and is implicated in Ca$^{2+}$ signaling regulation[5,6] and in immune cell development[7,8]. In particular, the A isoform of IP3K, which is located in brain and testis, is important for dendritic morphology[9], synaptic plasticity and impairments of learning and memory[10]. More recently, it has been shown that IP3KA is expressed ectopically in a broad range of tumor types, playing an important role for tumor growth and metastasis[11–13].

IP3K is organized in several domains, including an N-terminal domain responsible for protein localization, a Calmodulin (CAM) binding domain responsible for protein activation, and a C-terminal

[1]Department of Crystallography and Structural Biology, Institute of Physical-Chemistry Blas Cabrera, CSIC, Serrano 119, 28006 Madrid, Spain. [2]School of Biological Sciences, University of East Anglia, Norwich Research Park, Norwich NR4 7TJ, UK. [3]Drug Discovery and Medicinal Chemistry, Department of Pharmacology, University of Oxford, Mansfield Road, Oxford OX1 3QT, UK. [4]Wolfson Laboratory of Medicinal Chemistry, Department of Life Sciences, University of Bath, Claverton Down, Bath BA2 7AY, UK. [5]These authors contributed equally: Hayley Whitfield, Andrew M. Riley. [6]Deceased: Shane W. Garrett. ✉e-mail: barry.potter@pharm.ox.ac.uk; xbeatriz@iqf.csic.es

catalytic domain with the kinase catalytic function. The N-terminal domain of the A and B isoforms comprises an actin binding domain that mediates localization to F-actin[14,15], which is important for regulation of actin dynamics[15]. Currently, for IP3KA, only structures of the catalytic and CAM binding domains are available[16–18]. The structure of the IP3K kinase domain (IP3K-KD) revealed that the inositol polyphosphate kinase (IPK) family conserves the protein kinase (PK) fold organization in two lobes, the N- and C-lobes, binding the ATP in between them in a similar way. However, they present a third lobe, the IP-lobe inserted into the C-lobe, which accounts for most of the interactions with the inoside substrate. Available structures to date provide a full structural characterization regarding substrate (InsP₃) and product (InsP₄) recognition, revealing why IP3K is such a specific enzyme for InsP₃ and also for the position of phosphorylation, the secondary hydroxyl group at position 3 (3-OH). From the IPK family, IP3K shows the greatest substrate specificity and selectivity, whereas inositol polyphosphate multikinase (IPMK) displays a great promiscuity, phosphorylating various InsP₃ and InsP₄ isomers at different positions. The shape of the IPKs IP-lobe is related to this aspect, IPMK exhibiting the smallest IP-lobe with just one helix[19] whereas IP3K presents four helices that generate a more constrained active site[16]. InsP kinases from other families also contrast with IP3K high specificity, as is the case of the ITPK1 family of enzymes; however, this family displays a different fold that could account for this[20].

Both IP3K activities, catalytic and F-actin bundling, have been linked to the oncogenic potential of IP3KA[11]. In support, inhibition of catalytic IP3K activity reduces proliferation and adhesion of lung cancer cells, diminishing their metastatic potential[21,22]. Inhibition based on the ATP binding site has been explored using purine derivatives[23]. In addition, some of us have explored targeting the InsP₃ binding site based upon InsP₃ analogues[24]. The chemistry of both natural inositol phosphates and their analogues has been thoroughly reviewed[25,26]. Finally, structurally more distant compounds were identified by high throughput screening, with IC₅₀ values in the nM range[22]. Other targets, directed at the IP₃R and based upon the natural product adenophostins have also been addressed, a recent example being a synthetic inositol ribophostin[27–29].

Despite all this, it is not wholly clear what departures from the structure of InsP₃ might be compatible with binding and even catalytic activity. Here, a diverse range of ligands related to InsP₃ have been interrogated for their interaction with IP3K and ability to act as IP3K substrates using chemical synthesis, protein crystallography, HPLC analysis, molecular docking, thermal shift experiments and fluorescence polarization of a synthetic fluorescently-tagged FITC-InsP₃. We have surveyed a subset of synthetic InsP₃ and InsP₄ analogues that includes both chemical variations in several positions of the inositol ring with respect to the natural substrate, including the critical 3-position[28–36] and even related compounds based upon a non-cyclitol

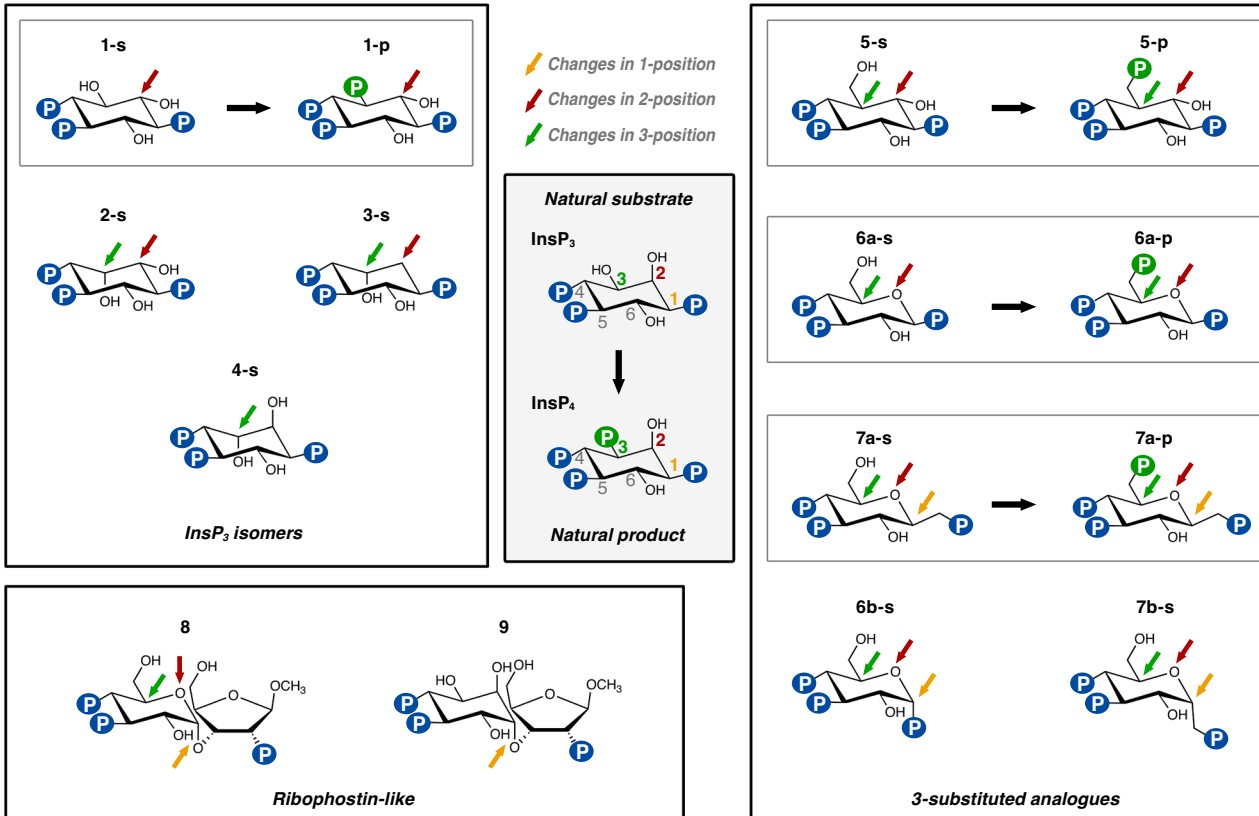

**Fig. 1 | 2D representation of InsP mimics.** The central square represents the natural IP3K substrate (InsP₃) and product (InsP₄). The remaining compounds are mimics of substrates (-s) or products (-p) used in our experiments with the exception of compounds **6a–p** and **7a–p**, which were found in IP3K complexes generated from **6a–s** and **7a–s** respectively. We have classified these compounds into three groups: InsP₃ (or InsP₄) isomers or analogues (**1–4**); InsP₃ (or InsP₄) analogues with a primary hydroxyl (CH₂-OH) at pseudo 3-position or carbohydrate equivalent (**5–7**) and ribophostin or analogues (**8–9**). Small arrows mark substituents that differ from InsP₃ colored as InsP₃ 1 (yellow), 2 (red) and 3 (green) positions. Inner-squares show pairs of substrate/product mimics. The symbol of a P in a circle represents a phosphate group, with the phosphates transferred by phosphorylation (at pseudo 3-position) shown in green. The following specifies the nomenclature for compounds and the isomers used in this work: **1-s** (L-*scyllo*-inositol 1,2,4-trisphosphate), **1-p** (*scyllo*-inositol 1,2,3,5-tetrakisphosphate), **2-s** (D-*myo*-inositol 1,4,6-trisphosphate), **3-s** (D-3-deoxy-*myo*-inositol 1,4,6-trisphosphate), **4-s** (L-*chiro*-inositol 2,3,5-trisphosphate), **5-s** (DL–6-deoxy-6-hydroxymethyl-*scyllo*-inositol 1,2,4-trisphosphate), **5-p** (DL-6-deoxy-6-phosphoryloxymethyl-*scyllo*-inositol 1,2,4-trisphosphate), **6a–s** (β-D-glucopyranosyl 1,3,4-trisphosphate), **6a–p** (β-D-glucopyranosyl 1,3,4,6-tetrakisphosphate), **7a–s** (β-D-glucopyranosylmethanol 3,4,1'-trisphosphate), **7a–p** (β-D-glucopyranosylmethanol 3,4,6,1'-tetrakisphosphate), **6b–s** (α-D-glucopyranosyl 1,3,4-trisphosphate), **7b–s** (α-D-glucopyranosylmethanol 3,4,1'- trisphosphate), **8** (ribophostin) and **9** (D-*chiro*-inositol ribophostin).

**Fig. 2 | Syntheses of *scyllo*-inositol 1,2,3,5-tetrakisphosphate (1-p), DL-6-deoxy-6-hydroxymethyl-*scyllo*-inositol 1,2,4-trisphosphate (racemic 5-s) and DL−6-deoxy-6-phosphoryloxymethyl-*scyllo*-inositol 1,2,4-trisphosphate (racemic 5-p).** Reagents and conditions: (**a**) NaBH$_4$/MeOH/THF, 89%; (**b**) i. 1 M HCl/MeOH 1:10, reflux; ii. conc. aqueous NH$_3$, 68%; (**c**) i. (CEO)$_2$PNPr$^i_2$, 1*H*-tetrazole, CH$_2$Cl$_2$, ii. *m*-CPBA, −78 °C, 85%; (**d**) Na/liq NH$_3$, −78 °C, 71%; (**e**) Me$_3$NBH$_3$, AlCl$_3$, 4 Å sieves, THF,

0 °C, 23 h, 65%; (**f**) 1 M HCl/EtOH 1:2, reflux, 87%; (**g**) i. (BnO)$_2$PNPr$^i_2$, 1*H*-tetrazole, CH$_2$Cl$_2$, ii. *m*-CPBA, −78 °C, 85%; (**h**) Na/liq NH$_3$, −78 °C, 71%; (**i**) 1 M HCl/EtOH 1:2, reflux, 71%; (**j**) i. (BnO)$_2$PNPr$^i_2$, 1*H*-tetrazole, CH$_2$Cl$_2$, ii. *m*-CPBA, −78 °C, 72%; (**k**) H$_2$, Pd-C, MeOH, 40 p.s.i., 75%. Bn = benzyl, PMB = *p*-methoxybenzyl, CE = 2-cyanoethyl. All asymmetrical compounds are racemic.

carbohydrate core. Our results shed light on IP3K active site plasticity and tolerance to different ligands and, of particular interest, on the surprising atypical IP3K phosphorylation of unexpected substrates. This work also defines unexpected IP3K biosynthetic capacities yielding innovative phosphorylated InsP-derived compounds, as well as providing a structural basis to aid future design of molecules of potential biomedical value.

## Results

### Synthesis of inositol phosphate ligands

We have used several InsP$_3$ and InsP$_4$ analogues (Fig. 1 and Supplementary Fig. 1) to analyze their experimental binding to IP3KA kinase domain (IP3K-KD). Syntheses of a number of these have been previously reported, as listed in the Methods Section, but three new analogues are employed in this study, *scyllo*-inositol 1,2,3,5-tetrakisphosphate (**1-p**), DL-6-deoxy-6-hydroxymethyl-*scyllo*-inositol 1,2,4-trisphosphate (**5-s**) and its phosphorylated product DL−6-deoxy-6-phosphoryloxymethyl-*scyllo*-inositol 1,2,4-trisphosphate (**5-p**). Synthetic routes are presented above (Fig. 2) and *vide infra*. [Note that for compounds **5-s** and **5-p**, used as racemates, only the L-enantiomers are shown in Fig. 1. These are the ones with phosphate regiochemistry directly comparable to InsP$_3$ and that are selected by the enzyme. In addition to the standard lowest locant numbering used previously and in Fig. 1, the analogues **5-s** and **5-p** could also be formally named as D-3-deoxy-3-hydroxymethyl-*scyllo*-inositol 1,4,5-trisphosphate and D-3-deoxy-3-phosphoryloxymethyl-*scyllo*-inositol 1,4,5-trisphosphate, respectively, to make the comparison to the D-enantiomers of InsP$_3$ and InsP$_4$ more readily appreciated].

### Synthesis of *scyllo*-inositol 1,2,3,5-tetrakisphosphate (1-p).

*Scyllo*-Ins(1,2,3,5)P$_4$ (**1-p**) possesses a plane of symmetry and therefore exists as a single *meso*-compound rather than as a pair of enantiomers. This makes it a particularly accessible analogue of Ins(1,3,4,5)P$_4$ synthetically, in that neither a stereospecific synthesis nor an optical resolution is required. The synthesis (Fig. 2) began with the versatile 2-inosose (**12**)[37], which was reduced rapidly and stereospecifically with sodium

borohydride to the *scyllo*-inositol orthoformate derivative **13**. Selective cleavage of the orthoformate ester gave tetrol **14**. Phosphitylation with bis(2-cyanoethoxy)diisopropylaminophosphine in the presence of 1*H*-tetrazole, followed by oxidation of the intermediate tetrakisphosphite triester with 3-chloroperoxybenzoic acid (*m*-CPBA) gave fully-protected **15**. Finally, removal of all protecting groups using sodium in liquid ammonia provided **1-p**, which was purified by ion-exchange chromatography on Q Sepharose Fast Flow resin and isolated as the triethylammonium salt in 37% overall yield from **12**.

### Synthesis of DL-6-deoxy-6-hydroxymethyl-*scyllo*-inositol 1,2,4-trisphosphate (racemic 5-s).

Racemic **5-s** was synthesized in four steps (Fig. 2) from the key racemic intermediate **16**[37] (Supplementary Methods and Supplementary Fig. 2). Thus, regioselective reduction of the benzylidene acetal in **16** using borane-trimethylamine complex and aluminium chloride in the presence of 4 Å molecular sieves gave the alcohol **17** in 65% yield. The two *p*-methoxybenzyl protecting groups of **17** were removed by acid hydrolysis giving triol **18**, which was phosphitylated using bis(benzyloxy)diisopropylaminophosphine/1*H*-tetrazole. The $^{31}$P NMR spectrum of the intermediate trisphosphite triester showed a large $^5J_{PP}$ coupling of 6.1 Hz between the phosphorus atoms of the vicinal phosphite groups (values of 3 or 4 Hz are typical for vicinal phosphites in an inositol ring). Oxidation with *m*-CPBA gave the fully protected trisphosphate triester **19** that was a low-melting point crystalline solid. Deprotection using sodium in liquid ammonia, and purification by ion exchange chromatography as for **1-p** went smoothly and the racemic **5-s** was obtained as the pure triethylammonium salt.

### Synthesis of DL-6-deoxy-6-phosphoryloxymethyl-*scyllo*-inositol 1,2,4-trisphosphate (racemic 5-p).

Racemic **5-p** was also synthesized from intermediate **16** (Fig. 2) that was re-synthesized for this purpose essentially according to Riley et al.[37]. The benzylidene acetal and the two *p*-methoxybenzyl groups in **16** were removed together in one pot by treatment with refluxing ethanol/1 M aqueous HCl (1:2), giving tetrol **20** in 71% yield after chromatography. Phosphitylation of **20** with

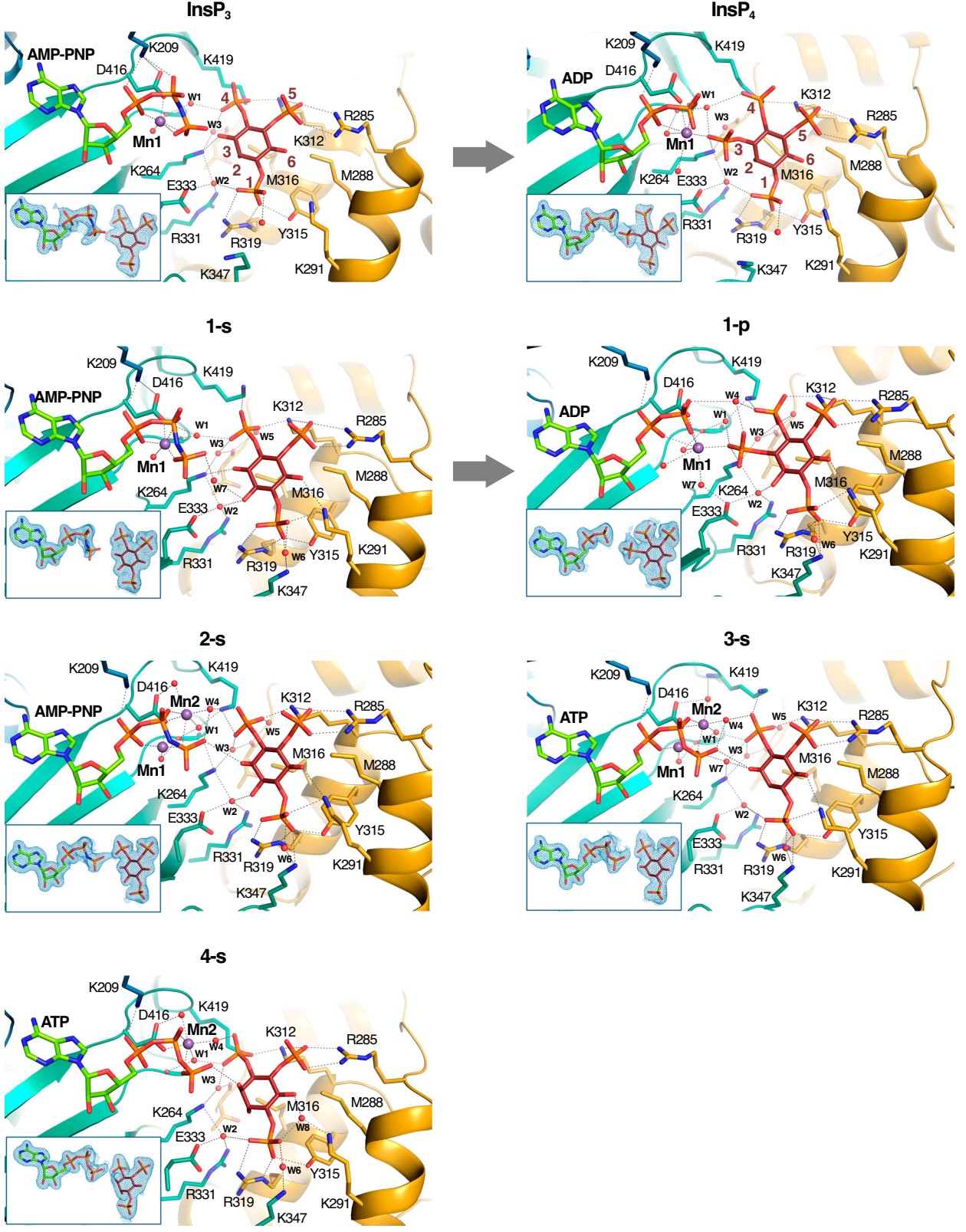

**Fig. 3 | Crystal structures of IP3K-ligand complexes.** A zoom of IP3K-ligand structures showing the ligand recognition site for InsP₃ and InsP₄ isomers (**1–4**). At the top of the figure the complexes with substrate InsP₃ and product InsP₄ properly numbered are shown to facilitate comparison (PDB codes 1w2c and 1w2d respectively). Protein is represented as cartoons showing important residues as sticks (N-lobe in blue, C-lobe in green and IP-lobe in gold), ligands as sticks (carbons in lemon for nucleotide and brown for inositide, oxygens in red, nitrogen in blue and phosphates in orange) and ions and waters as purple and red spheres respectively. Arrows link IP3K-substrate complexes with their related IP3K-product complexes.

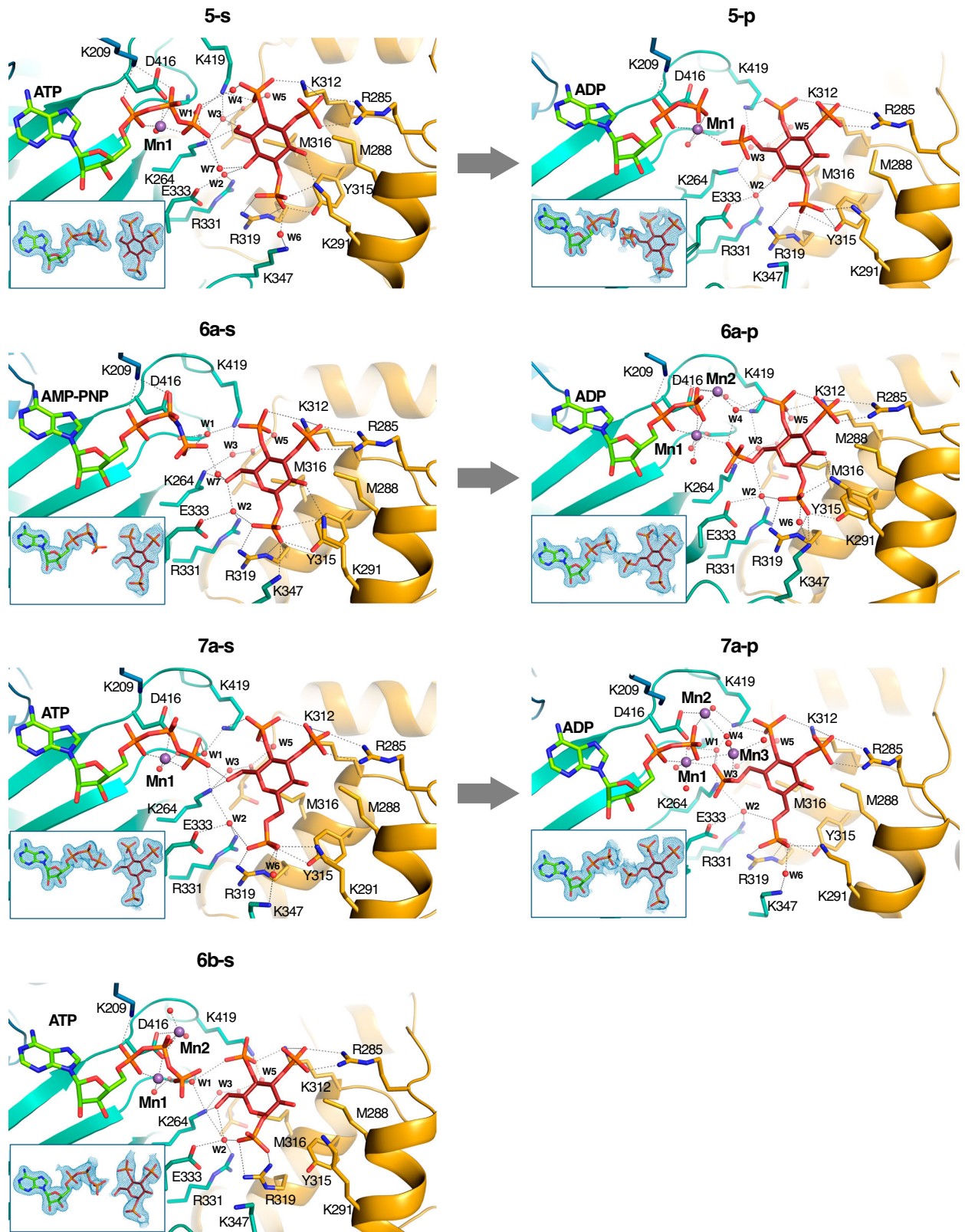

**Fig. 4 | Crystal structures of IP3K-ligand complexes.** A zoom of IP3K-ligand structures showing the ligand recognition site in 3-substituted analogues (**5**–**7**). Color code is the same as Fig. 3. Arrows link IP3K-substrate complexes with their related IP3K-product complexes. The IP3K complex with the expected product from the atypical substrate **5-s** has been formed by crystal soaking with **5-p**. IP3K crystal complexes with products **6a**–**p** and **7a**–**p** have been obtained by crystal soaking with the substrates **6a**–**s** and **7a**–**s**.

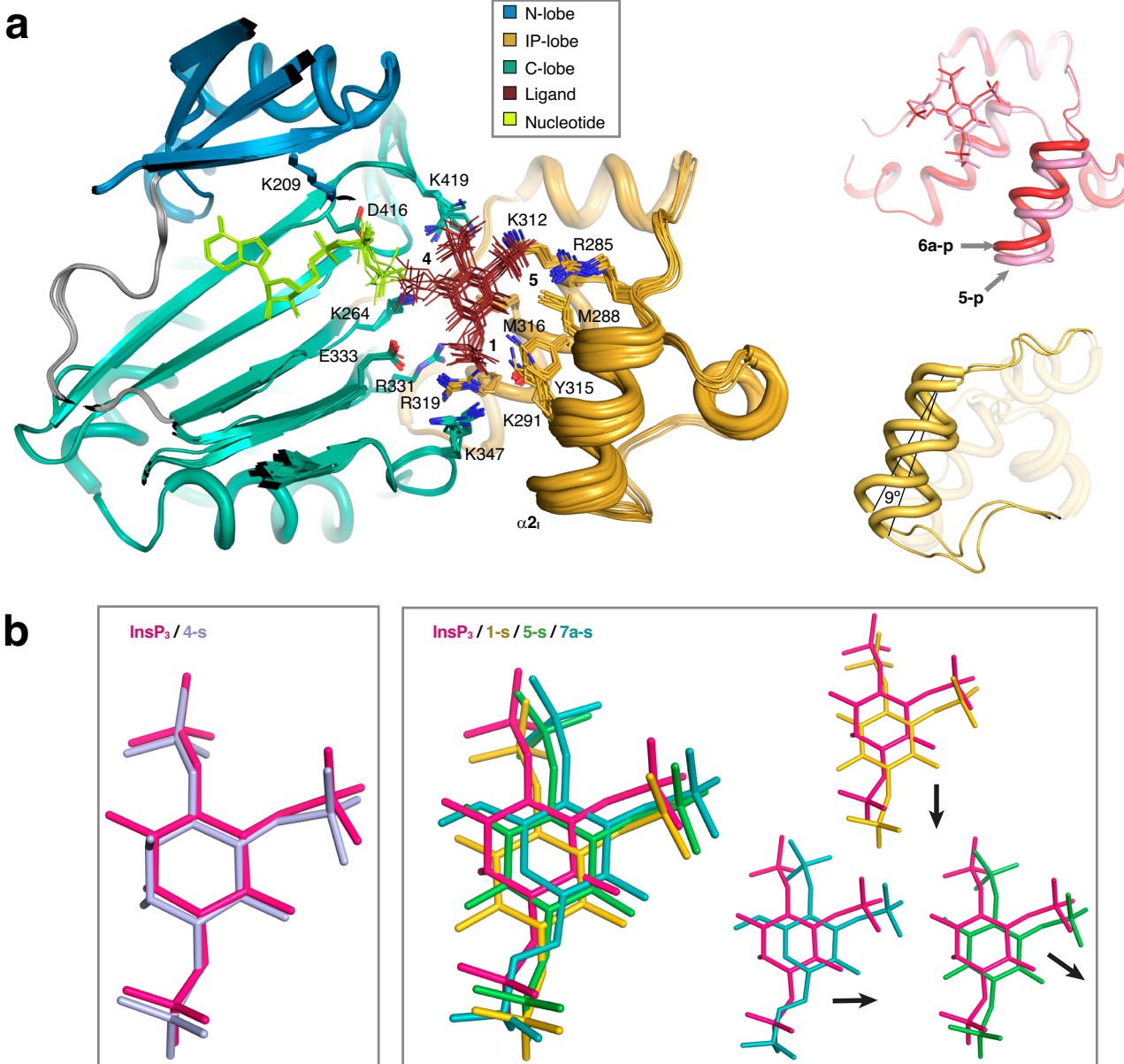

**Fig. 5 | The IP3K IP-lobe spring effect and InsP site variations. a** Structural superposition of the twelve IP3K-ligand complexes obtained in this work including IP3K/InsP₃ (code 1w2c) and IP3K/InsP₄ (code 1w2d) complexes. IP3K structures are shown as cartoons highlighting the three lobes in different colours (colour code as detailed in Fig. 3). The main residues for ligand recognition are numbered and shown as sticks. InsPs and analogues are shown as brown thin sticks. On the right, two pictures of the IP-lobe from the IP3K complexes with **6a-p** (red) and **5-p** (pink colored), showing the maximum shift of the IP-lobe helix α2ₗ. This shift is the spring effect referred to in the text, which consists of an helix-tilt of 9°. **b** Schematic representation of ligand positions superposed to the InsP₃ position. InsP₃ site is shown by InsP₃ and **4-s** harboring an axial OH in pseudo 2-position (left). By contrast, three types of deviations from the InsP₃ site are found in the studied analogues represented by **1-s** (compounds in this group: **1-s, 2-s, 3-s, 6a−s**), **5-s** (compounds in this group: **5-s, 6b−s**) and **7a−s** (right)). The small pictures show a superposition one by one displaying the shift direction by the arrows. Each ligand is colored according to the abbreviation at the top of its panel.

bis(benzyloxy)diisopropylaminophosphine/1*H*-tetrazole in dichloromethane gave a P(III) tetrakisphosphite that was not isolated, but the reaction mixture was checked by ³¹P NMR spectroscopy and signals observed at δ140.8 (doublet), 142.3 (singlet), 142.5 (doublet), and 143.0 (singlet). Immediate oxidation with excess *m*-CPBA at −78 °C, then purification of P(V) product by column chromatography gave the desired benzyl-protected tetrakisphosphate **21** in 72% yield. Deprotection was achieved by catalytic hydrogenolysis using palladium on charcoal in MeOH/water (4:1) to yield the desired tetrakisphosphate that was purified by ion-exchange chromatography on Q Sepharose Fast Flow resin, eluting with a gradient of triethylammonium bicarbonate buffer to yield racemic **5-p** as the triethylammonium salt in 75% yield.

## IP3K recognizes a variety of InsP₃ mimics

We have obtained multiple crystals of IP3K-KD in the presence of most substrate (**-s**) and product (**-p**) mimics, as shown in Fig. 1. Compound numbering is shown in Supplementary Fig. 1. However, in this work we will also refer to their pseudo 1, 2, 3, 4, 5 or 6 positions, which are spatially equivalent to the 1, 2, 3, 4, 5 or 6 positions of InsP₃ (Fig. 1) in their binding mode relative to natural InsP₃. InsP₃/InsP₄ mimics can be classified into three groups regarding their chemical structure: (a) InsP₃ (or InsP₄) isomers or analogues (**1–4**); (b) 3-substituted InsP₃ (or InsP₄) analogues with a primary hydroxyl group (-CH₂-OH) at the *myo*-inositol pseudo 3-position (**5–7**) and (c) ribophostin or analogues (**8–9**). Statistics

for crystal data processing and data refinement are shown in Supplementary Table 1.

We observe overall that ligand binding is very conserved with that of InsP₃ (Figs. 3, 4). Summarizing, IP3K substrate recognition entails close interactions with the three phosphates in positions 1, 4 and 5 of the InsP or of the corresponding carbohydrate-based analogues. Regarding the hydroxyls, the key 3-OH is recognized by Lys264 through hydrogen bonds, whereas no direct interactions are produced with OH groups in positions 2 and 6, though they keep van der Waals interactions with Met316 and Met288 respectively. In some of the complexes we observed the participation of two basic residues, Lys291 and Lys347, in a second sphere of InsP₃ recognition. Interestingly, changing the natural InsP₃ ring substituents leads to IP-lobe adjustments resulting in an helix tilt up to 9° in α2₁ generated by the contraction of the hinges connecting this helix to the protein (Fig. 5a, Supplementary Video 1). From now on we will refer to this motion as a spring effect. This effect on the IP-lobe, together with side chain adjustments, provides a plastic active site able to accommodate different ligands (Supplementary Fig. 3a–c).

The above spring effect is coupled with a noticeable shift of the ligands away from the InsP₃-site (Fig. 5b). We observed that this depends on the nature of the substituent at pseudo 2-position of the inositol ring. Thus, an axial OH in this position, (InsP₃ and compound **4-s**), forces the ligand to fit in the InsP₃-site (Fig. 5b). However, all other InsP₃ mimics used, with either an equatorial OH or no substituent at this pseudo 2-position, show a noticeable shift from the InsP₃-site in different directions, depending on the analogue (Fig. 5b). Probably, either the van de Waals interactions produced between the axial 2-OH and Met316 or conserved water-mediated interactions between the equatorial 2-OH and three protein residues (Lys264, Arg331 and Glu333) could account for variations in the compounds' binding relative to InsP₃ (Figs. 3, 4).

After the catalytic reaction, the hydroxyl at position D−3 of the D-*myo*-inositol ring is phosphorylated and the protein maintains the ability to bind the product, at least at our experimental concentrations. Some of the compounds analyzed, such as **1-p, 5-p, 6a−p** and **7a−p**, are in fact mimics of the natural product InsP₄ (Figs. 1, 3, 4). The complexes with **1-p** and **5-p** were obtained by soaking crystals directly with these product analogues. In contrast, complexes with **6a−p** and **7a−p** were obtained when crystals were soaked into solutions containing the substrate mimics **6a−s** and **7a−s** under specific conditions (see below). Substrate/product pairs only suffer a slight readjustment of the inositol polyphosphate position in the IP3K active site (Supplementary Fig. 3d). Otherwise, binding of product mimics obeys the same rules as before.

Finally, as with InsP₃ itself, nearly all the complexes with substrate mimics keep the Mn²⁺ ion (Mn1) that bridges the ATP phosphates with the enzyme and orients Pγ for catalysis. Furthermore, we found up to three metal positions depending on the complex. Interestingly, complexes of IP3K with products formed in situ in the crystallization experiment exhibit two (**6a−p**) and three (**7a−p**) metal positions (Supplementary Fig. 4). The role of metal positions seems to be key in ATP binding, Pγ orientation, and transition state and product stabilization.

### Improving IP3K ligands

Most ligands shown in Fig. 1 were submitted to ligand and thermal shift assays (Fig. 6). Noting that 2-FITC-InsP₃ (Fig. 6a) proved a powerful ligand of the InsP₃ binding site of an InsP₃ receptor[38], we tested this compound for binding to IP3K-KD under conditions with inclusion of EDTA, which prevents catalytic processing of the compound (Supplementary Methods and Supplementary Fig. 5). Curves for displacement of 2-FITC-InsP₃ by diverse substrates, analogues and their products are shown, with derived IC₅₀ values, in Fig. 6a and summarized in Supplementary Table 2. Thermal shift (denaturation) curves for

IP3K-KD in the absence or presence of ligands are shown in Fig. 6b. Derived values of temperature of unfolding and its variation upon ligand binding are shown in Fig. 6b and Fig. 6c respectively and in Supplementary Table 3. Note that measurements from chemical and thermal shift experiments show high correlation (values of IC₅₀ from Fig. 6a, as well as inferred K_d values, are shown as a bar at the bottom of Fig. 6c). We found that most compounds stabilize the enzyme significantly with the exception of ribophostin (**8**) and its analogue (**9**) (Fig. 6c). Note that compounds derived from the phosphorylated α-glucose anomer (**6b−s**) and the extended α-*C*-glycoside (**7b−s**) as InsP₃ mimics stabilize IP3K-KD much less than InsP₃ (Fig. 6c), probably due to the loss of interaction with Tyr315 (Fig. 4). Notably, it has not been possible to get crystal complexes with them, with the exception of compound **6b−s**, that surprisingly was captured in the crystal.

All InsP₃ (or InsP₄) isomers (compounds **1**−**4**) retain the ability to stabilize the enzyme significantly. We observe that the *scyllo* analogues of InsP₃ and InsP₄ (**1-s** and **1-p**) with an inverted pseudo 2-OH group (therefore with all substituents equatorial) are better ligands for IP3K than the natural ones, exhibiting IC₅₀ values of 36 nM and 51 nM respectively, compared to a value of 140 nM for InsP₃. After these, the best compounds are those with just one substituent axial, either at pseudo 2 or 3 positions (InsP₃ and compound **2-s**). Finally, the presence of two axial -OH groups (**4-s**) constitutes the least stabilizing isomers in this group. Note that compound **3-s**, with one axial -OH and the absence of a substituent at pseudo 2-position, present an equivalent effect to the latter one.

Analysis of the 3-substituted InsP₃ analogues (compounds **5**−**7**) revealed that a change in the pseudo 3-position substituent from a secondary and equatorial alcohol (-CHOH) as in natural InsP₃, to a primary, but still equatorial, hydroxyl group (-CHCH₂-OH), leads to a slight decrease in protein stabilization, although the analog still retains the ability to stabilize the enzyme significantly (Fig. 6c). The effect on stabilization is similar in the compounds derived from the *scyllo*-inositol template (**5-s**) and β-D-glucose template (**6a−s**). However, IC₅₀ measurements indicate that the β-D-glucose derivatives (IC₅₀ 205 nM) are better ligands than the *scyllo* derivatives (IC₅₀ 972 nM), suggesting that a pyranoside oxygen placed at the pseudo 2-position is a promising feature (Fig. 1). Nevertheless, note that the *scyllo* derivatives (**5-s** and **5-p**) are synthetic racemates and, due to the fact that IP3K only selects the L-isomer (*vide supra* for discussion of numbering), the IC₅₀ measurements of this compound could likely be underestimated by a factor of two. The effects of the mirror image D-isomer, not selected in both crystals, are unknown but we presume they possess little or no binding activity.

Compounds of these series of 3-substituted analogues harbor different substituents at the pseudo 1-position. Analysis revealed that analogues derived from the α-glucose polyphosphate anomer (**6b−s**, IC₅₀ 8977 nM) and the corresponding C-glycoside (**7b−s**, IC₅₀ 6986 nM), i.e. with an axial pseudo 1-phosphate (1-P) instead of an equatorial one, barely stabilize the enzyme (Fig. 6). However, extending the phosphate position by changing the natural P substituent to a C-glycoside CH₂-P group, but keeping the equatorial configuration (see **7a−s**, IC₅₀ 656 nM, *vs* **6a−s**, IC₅₀ 205 nM), results in a worse ligand by IC₅₀ measurements, though still retaining a good ability to stabilize the enzyme.

Finally, product InsP₄ and *scyllo*-analogues (**1-p**, IC₅₀ 51 nM, and **5-p**, IC₅₀ 875 nM) behave similarly to their respective substrate and analogues, explaining why IP3K can easily be crystallized in the presence of products, at least at saturating concentrations.

In summary, changing the pseudo 2-OH configuration from axial to equatorial increases binding capability, whereas changing the configuration of other substrate substituents from equatorial to axial slightly decreases it. If the change of an equatorial group to an axial one involves a phosphate group, however, this decrease is dramatic. In a similar way, enlarging the -OH group and the phosphate groups at

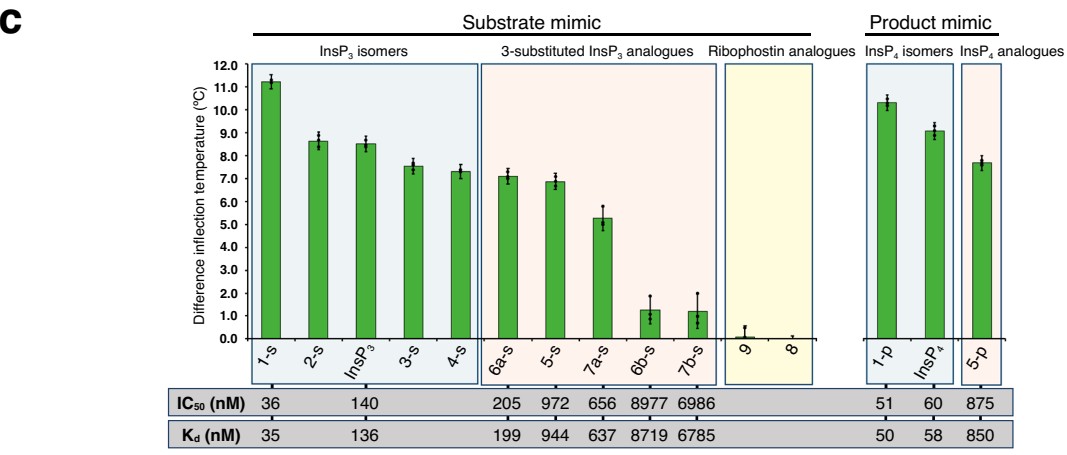

**Fig. 6 | Evaluation of IP3K ligand binding. a** Displacement of 2-FITC-InsP₃ from IP3K-KD by substrate and analogues, showing the structure of 2-FITC-InsP₃. **b** Examples of thermal denaturation curves for apo IP3K-KD (without ligands) and IP3K-KD in the presence of selected InsP analogues. The graph on the right shows the first derivative of the curves. The colors for each ligand curve are indicated in the middle square. The dots on both graphs represent the $T_i$ values of each experiment. The table below shows the mean of inflection temperature ($T_i$) values for the first transition of each curve, which show larger variations than the second transition. **c** Bar representation of mean $\Delta T_i$, showing the three individual points, in

the first transition experimented by IP3K-KD upon each ligand binding. IC₅₀ (and $K_d$) values are shown at the bottom to facilitate comparison. Error bars in (**a**) show the mean and 95% confidence interval from $n = 4$ discrete measurements of anisotropy of a single sample at the given protein: displacing ligand ratio. All samples were measured in a 384 well plate besides InsP₃, which in two separate experiments (performed on separate occasions) gave IC₅₀ values of 140.1 and 139.3 nM (data in Supplementary Table 2). The mean of $T_i$ measurements from $n = 3$ and the respective SDs are represented in table (**b**) and the graph (**c**, error bars) (data in Supplementary Table 3). Source data are provided as a Source Data file.

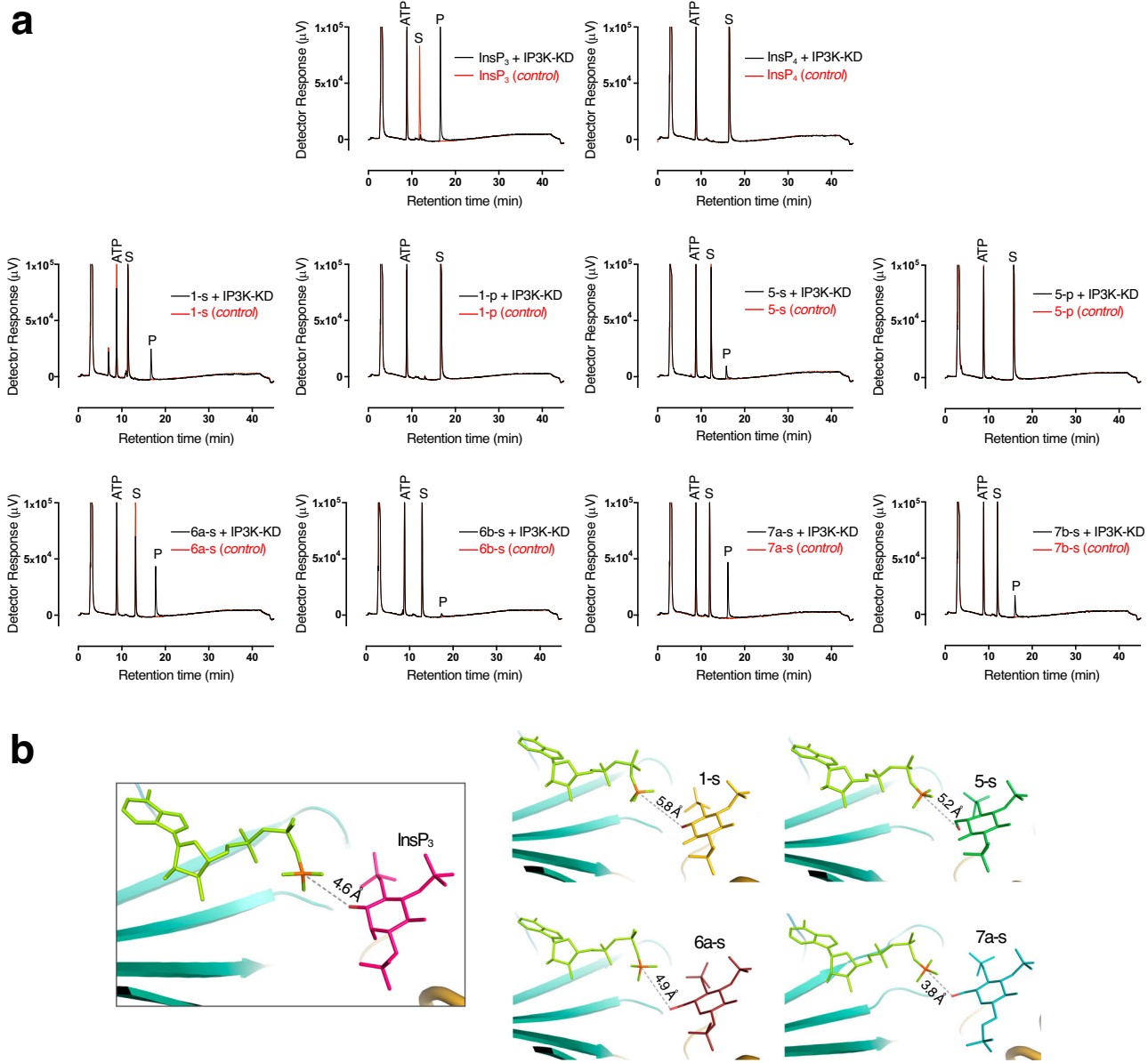

**Fig. 7 | Unexpected IP3K substrates. a** HPLC resolution of reaction products of assay of IP3K-KD with InsP analogues. With IP3K (black line), without IP3K (red line). ATP elutes at 8 min. InsP analogues and products obtained in the reaction thereof elute after 10 min. InsP analogues tested as substrates are indicated S, products of the reaction obtained are indicated P. **b** A zoom of the IP3K active site in complex with its natural substrate (left) and unexpected substrates (four pictures in the right); showing the distance between the reactive atoms (Pγ and O on 3-position - *myo*-inositol numbering) highlighted in orange (P) and red (O). Source data are provided as a Source Data file.

pseudo 3 and 1-positions by $CH_2$-OH and $CH_2$-P groups respectively, slightly decreases the binding capability, though to a significantly greater extent in the latter.

## IP3K atypical phosphorylation on primary hydroxyls

The *scyllo*-InsP₃ analogues (**1-s** and **1-p**) as well as pseudo 3-substituted analogues (**5-s, 5-p, 6a−s, 6b−s, 7a−s** and **7b−s**) were examined as IP3K substrates. HPLC analysis of reaction mixtures was performed to assess progress of the enzymatic reaction (Fig. 7a). We found that ʟ-*scyllo*-1,2,4-trisphosphate (**1-s**) is a good IP3K substrate. In further support, the crystal structures in complex with substrate **1-s** and its putative product, the synthetic symmetrical *scyllo*-inositol 1,2,3,5-tetrakisphosphate (**1-p**, in this study, equivalent to natural InsP₄ in *myo*-inositol polyphosphate regiochemistry), show that they both perfectly mimic InsP₃ and InsP₄ binding (Fig. 3). The main difference is the shift from the InsP₃ site in the *scyllo*-derived compounds (Fig. 5b). This shift causes a distance increase between the Pγ of ATP and the pseudo 3-OH of the inositol substrate by more than 1 Å (from 4.6 Å in the InsP₃ complex to 5.8 Å in that for **1-s**) (Fig. 7b). This confirms that a shift from the InsP₃-site, and therefore a slight increase in the distance between reactive positions, is still compatible with catalysis.

Unexpectedly, the InsP₃ analogues substituted at the pseudo 3-position with primary hydroxyls are also substrates, as shown by HPLC analysis on reaction mixtures using compounds **5-s, 6a−s**, and **7a−s**, including their pseudo 1-axial analogues **6b−s** and **7b−s**, though the latter seem to be weaker substrates than the former (Fig. 7a). Since these compounds expand the putative reactive position by the one carbon-carbon covalent bond, this surprising finding was further characterized by X-ray crystallography.

Thus, we obtained crystal complexes of IP3K-KD with the atypical substrates **5-s**, **6a–s**, **6b–s** and **7a–s** (Fig. 4). In all cases, the primary hydroxyl occupies a similar position to the 3-OH from InsP₃ remaining close to the ATP Pγ (Figs. 4, 7b). It should be noted that the synthetic **5-p** used as a HPLC standard and **5-s** compounds (Figs. 4, 7a) were racemic in this study, but only one enantiomer of **5-s**, that mimicking the phosphate regiochemistry of D-InsP₃, is assumed to interact with the enzyme to give the unique product **5-p**. In concordance, the crystal structure of the IP3K-KD/**5-p** complex shows that **5-p** binds mimicking the InsP₄ phosphate regiochemistry (Fig. 4). Furthermore, HPLC with authentic, albeit racemic, synthetic standard shows that compound **5-p** and the product of the reaction of compound **5-s** coelute (Fig. 7a). We also explored different soaking conditions for IP3K-KD crystals in presence of ATP/Mn²⁺ plus **6a–s** or **7a–s** compounds shown to be substrates by HPLC (Fig. 7a). Increasing the ratio nucleotide/inositide enough to still capture the latter allowed the reaction to proceed and the crystals to capture the products of the reaction **6a–p** and **7a–p** (Fig. 4). These data reveal that phosphorylation unambiguously also occurs at the primary 6-OH hydroxyl group (carbohydrate numbering) i.e. also at a group placed at the pseudo 3-position (*myo*-inositol numbering) of the two atypical and chiral carbohydrate-based substrate mimics of InsP₃.

These results represent a demonstration that IP3K, or an IPK, can not only phosphorylate more diverse compounds than those based on cyclitols. i.e. those based upon a carbohydrate template, but also at their primary hydroxyl group when they possess the appropriate regiochemistry of polyphosphate substitution. These data open up singular potential to harness the biosynthetic ability of this enzyme.

## Docking of 3-substituted *myo*-InsP analogues and substrate ability

To identify further IP3K ligands and substrates virtually, we have validated for our system two docking strategies using the GOLD program, as described in Methods. Briefly, a first approach docking each ligand against its experimental protein structure and a second one, named ensemble docking, using all experimental protein structures instead. Both strategies yield solutions that approximately reproduce the overall aspects from the experimental results obtained from the IP3K-KD complexes, but only in the case of complexes with substrate mimics. Binding of product and its analogues (Supplementary Fig. 6a) was not predicted. It's noteworthy that the ensemble docking, which introduces protein flexibility employing a repertoire of obtained experimental structures, improves the docking results compared to using only the previously available structure of IP3KA-KD in complex with InsP₃ (code 1w2c) (see Supplementary Fig. 6b, c).

The predictive results using InsP₃ mimics prompted us to search for binding models for the three compounds that failed to form protein-ligand crystal complexes (**7b–s**, **8** and **9**). Thus, we obtained a model for the α–substituted carbohydrate-based **7b–s**, which demonstrated poor substrate activity for IP3K. We hypothesize that this compound might require a slightly constrained conformation to effectively bind to IP3K, as suggested by the analysis of bond torsions (Supplementary Fig. 7a). This observation could potentially rationalize its limited substrate activity, as depicted in Fig. 7a. In parallel, models obtained for the two ribophostin-based analogues (**8** and **9**) show high energy conformations as compared with CSD geometrical distributions, which is in keeping with our failure to obtain crystal complexes or observe protein stabilization (Supplementary Fig. 7a).

Next, we searched for binding models for another two InsP₃ analogues using ensemble docking. The first one is a more distant InsP₃ known synthetic isomer (**10**) which presents several axial substituents (Supplementary Fig. 7b). The docking prediction for compound **10** suggests a rotation from the InsP₃ position in order to properly allocate the phosphates. While we maintained the ring torsions rigid during the docking run, we have considered the two possible

energetically related chair conformations, however, we cannot rule out the possibility that other cyclohexane conformations such as twist-boat could be selected by the enzyme.

The second one is based on the 3-substituted ligands (*myo*-inositol numbering) studied that lack an axial pseudo 2-OH. As mentioned above, these ligands cause a shift in the IP3K active site with respect to InsP₃ and therefore allow enough room for the pseudo 3-CH₂-OH to be accommodated (Fig. 7b). Therefore, in order to predict the binding to IP3K of an atypical *myo*-inositol substrate harboring an axial 2-OH group, otherwise equivalent to **5-s**, we designed a compound (**11**) (Supplementary Fig. 7b) and submitted it to docking analysis. From the best solutions, we propose that this ligand could also bind to IP3K and, as in the case of **5-s**, keep an adequate space between the Pγ and the primary hydroxyl (Supplementary Fig. 7c) though it would be more constrained. Further experimental validation is required to confirm the precise binding site of this ligand and its ability to be an IP3K substrate. This compound, to the best of our knowledge, has yet to be chemically synthesized.

From our analysis we therefore suggest that a variety of InsP₃ mimics possessing an extended OH group at the pseudo 3-position could be IP3K substrates, as long as they can exploit the discrete IP3K active site plasticity. The possible extent of the 3-OH branching or other possible variations remains to be determined.

## Discussion

IP3K biology is crucial for Ca²⁺ signaling and actin dynamics; overexpression of the enzyme in non-natural tissues is associated with cancer processes and metastasis[11].The work presented here outlines unexpected IP3K ligands and catalytic activity on atypical substrates.

Regarding unexpected IP3K ligands, we show here that InsP₃ and InsP₄ analogues with discrete variations in pseudo 1, 2 and 3-positions are good IP3K ligands. In summary, equatorial hydroxyls of the inositol ring increase ligand affinity, although pyranose-based derivatives are also good ligands for IP3K. This is of particular interest given that such ligands can be synthesized generally in chiral form and any required protecting group chemistry is usually much easier, whereas inositol derivatives often need tiresome separation of enantiomers. It should be noted, however, that the enzyme selects one enantiomer with the required regiochemistry of phosphate substitution from a racemate, at least in our limited case here. We also observed that small changes at the pseudo 2-position of the InsP do not notably affect ligand binding or protein stability in agreement with previous observations[24], although affinity may be affected. However, moderately enlarging the substituent at the pseudo 1-position decreases ligand binding whereas enlarging the substituent at pseudo 3-position has a much lower effect. Finally, at saturating concentrations, the product InsP₄ and its analogues are good ligands for IP3K suggesting that the IPK family might be inhibited by its reaction products, as in the case of lipid kinases[39].

Binding different ligands to IP3K revealed a protein plasticity. Previous studies showed that the IP3K IP-lobe presents great flexibility, displaying a noticeable structural change upon InsP₃ binding (Supplementary Fig. 8)[16]. We show here that IP3K also displays a moderate protein plasticity centered in the α2₁ helix of the IP-lobe (Fig. 5a) that enables IP3K to bind different InsP₃ analogues. Protein flexibility and plasticity have also been observed in other members of the IPK family. Thus, mammalian InsP₅ 2-K undergoes a change from an open conformation to sequential closed conformations upon nucleotide and InsP₅ binding respectively[40]. Regarding IPMK, acting on several InsP₃ and InsP₄ isomers, we believe that its unique IP-lobe helix (equivalent to IP3K α4₁) might also show plasticity (Supplementary Fig. 8d), though structural analysis in the presence of different InsP mimics would confirm this. The related lipid kinase PI3K has also shown great plasticity in the ATP site which is a good target point for selective inhibitor development against the different PI3K isoforms[41]. Finally, this is also a repeated feature in protein kinases, where structural plasticity is of the

utmost importance for kinase regulation and activity, with different functional states showing distinct conformations[42]. Targeting specific kinase conformations is a strategy to obtain more selective inhibitors. Undoubtedly, IP3K plasticity could be also exploited to design InsP ligands specifically, including potential bi-substrate ligands that represent a successful approach in the PK field to develop molecules without off-target effects derived from targeting the conserved ATP binding[43].

An exciting achievement in the field would be discovery of molecules able to discriminate between all the InsP$_3$ recognizing proteins that cover crucial events such as Ca$^{2+}$ signaling, cell survival or cell growth. Despite the IP3K protein plasticity mentioned, ligands with voluminous substituents at the pseudo 1-position, such as the bicyclic disaccharide-based InsP$_3$ mimic ribophostin (**8**) or its inositol analogue (**9**), or even the *C*-glycoside InsP$_3$ mimic[44,45] **7b**−**s**, do not, or barely, bind IP3K as we show experimentally herein (Fig. 6b,c) and through computational studies (Supplementary Fig. 7a). In agreement, other ribophostin-related analogues have previously been found not to be IP3K ligands[24]. This is particularly interesting due to the fact they act as potent agonists of IP$_3$R[28]. Conveniently, IP$_3$R and IP3K do not share a similar ligand recognition site, IP$_3$Rs displaying a more open site for phosphate P1 recognition than IP3K, as observed in their crystal structures[16,46]. We can conclude here that ligands with large substituents at the pseudo 1-position can be designed to bind to IP$_3$R highly potently without affecting IP3K, which is something of particular interest. It may be harder though to achieve the reverse specificity. Nevertheless, from this study we can also expand previous observations regarding the impact of substituents on pseudo 2 or 3-positions in both InsP$_3$ binding proteins. Thus, compound **5-s**, synthesized and shown to be a good IP3K ligand in this work, is equipotent with InsP$_3$ itself in releasing $^{45}$Ca$^{2+}$ from permeabilized rabbit platelets[47]. Note that the structure of **5-s** differs from InsP$_3$ in the 2-hydroxyl (equatorial *vs* axial) and the 3-substituent (hydroxymethyl *vs* hydroxyl). Since the 2-hydroxyl group is not important for Ca$^{2+}$ release[26] or IP3K binding (this work and Poinas et al.[24]), we conclude that discrete enlargement of the substituent in pseudo 3-position does not yield selective ligands between both enzymes. However, increasing the substituent at the pseudo 3-position to a methylenephosphate group as in **5-p**, the enzymatic product of **5-s** and also synthesized in this work, abolishes the ability to stimulate cellular Ca$^{2+}$ release[48], as determined in permeabilized rat hepatocytes[30]. The activity of **5-s** and inactivity of **5-p** was expected, as **5-p** bears the same relationship to the natural InsP$_3$ metabolite InsP$_4$ which does not release calcium. Therefore, if **5-p** is considered to be an InsP$_3$ surrogate, its inactivity indicates that the steric bulk of a phosphate attached to the primary hydroxyl group at the 3-position is not tolerated by the receptor. In contrast, we have observed that **5-p** maintains quite good binding to IP3K (Fig. 6), altogether suggesting that exploring the substituent in (pseudo) 3-position can reverse the selection towards IP3K.

Regarding the unexpected IP3K catalytic activity, previous studies with InsP mimics revealed that an equatorial secondary alcohol OH group at the *myo*-inositol pseudo 3-position is necessary to allow IP3K catalysis[24]. Interestingly, we show here that IP3K ligands still retain the ability to be substrates when the length of this 3-substituent is increased by an additional -CH$_2$- motif. This represents a highly unexpected finding, considering the high specificity of binding and position of phosphorylation of natural InsP$_3$ substrate. To the best of our knowledge, IPKs, as well as other families of InsP kinases, phosphorylate biological substrates at secondary equatorial hydroxyl groups, with the exceptions of the InsP$_5$ 2-kinase and InsP$_6$ kinase families, which phosphorylates axial hydroxyls and equatorial phosphate groups respectively[49,50]. Interestingly in cells, protein kinases mostly phosphorylate three different residues that act as nucleophiles: Ser (primary hydroxyl), Thr (secondary hydroxyl) and Tyr (tertiary aromatic hydroxyl). In small molecule kinases that also keep an in-line

phosphoryl transfer mechanism (see below), we found examples of a variety of nucleophiles comprising not only primary and secondary alcohols, but also phosphate groups, guanidinium groups or acidic groups[51]. In this work we have proven that, like Ser/Thr PKs, IP3K and presumably other IPKs such as IPMK, are able to phosphorylate primary hydroxyls; however, we do not know if IPK phosphorylation could be expanded to other substrate types, as has been observed with other types of kinases. Perhaps some guided mutations could engineer proteins with surprising abilities.

From the structure, we can assume that the phosphoryl transfer reaction consists of an in-line attack by the nucleophilic -OH onto the Pγ of ATP[16,51]. How much the mechanism has associative or dissociative features is well discussed in the protein kinase field[52–54]. Some crystallographic studies support a dissociative mechanism, such as that with protein kinase A (PKA)[55] and with protein phosphatase *At*PFA-DSP1[54] that succeeded in identifying a metaphosphate-like intermediate close to a fully dissociative transition state. A theoretical analysis indicates that the reaction coordinate distance (i.e., the optimal distance between the entering oxygen and the phosphorus undergoing substitution) for a fully dissociative transition state should be ≥4.9 Å[56]. In this work, the identification of unusual substrates (InsP$_3$ and compounds **1-s**, **5-s**, **6a**−**s** and **7a**−**s**) that expand the reaction center's distance (Fig. 7b) from 4.6 Å (InsP$_3$) to a range of 3.8−5.8 Å suggests that IP3K phosphorylates a range of substrates possibly with a strong contribution of a dissociative mechanism.

In conclusion, we have identified different limits for IP3K specificity, as well as defined unpredicted biosynthetic capabilities for this enzyme relevant to InsP$_3$ signaling. Our structural study reveals a moderate IP3K plasticity that allowed us to identify potentially available areas in the IP3K active site that represent an opportunity to design unusual IP3K ligands and even unexpected IP3K substrates. We can define the following features for IP3K substrate tolerance, i.e. what an InsP$_3$ analogue must exhibit to be an IP3K substrate: (1) it must contain three phosphates at the pseudo 1,4 and 5-positions in a six-membered ring, although this does not necessarily have to be a cyclitol, (2) its substituents must be accommodated inside active site plasticity margins, some of them established in this work and (3) it should feature an -OH group at a 3- or a pseudo 3-position, either as a primary or secondary hydroxyl, as confirmed by HPLC methodology and in situ capture of products by crystallography. Ultimately, by exploiting IP3K structural features, this work opens a window for generation of unprecedented and potentially valuable InsP derivatives that might be useful in both related cell biology and chemistry fields. Moreover, these could evolve into selective molecules for InsP binding enzymes (i.e. IP3K *vs* IP3R or other IPKs), which could be used for disease therapies or more accurately delineate unique roles for these pleiotropic biomolecules by in vivo cellular studies.

## Methods

### Synthesis of inositol phosphate analogues

2-FITC-InsP$_3$ was synthesized as described[38] and used as its triethylammonium salt. Thin layer chromatography (TLC) was performed on precoated plates (Merck TLC aluminum sheets silica 60 F$_{254}$, Art no. 5554). The spots were visualized using phosphomolybdic acid, potassium permanganate, iodine or UV light. All evaporations were carried out under reduced pressure. Flash chromatography was carried out using silica gel (Merck Kieselgel 60, mesh size 0.040−0.063 mm). $^1$H and $^{13}$C NMR spectra were recorded on either JEOL GX270 or EX400 spectrometers. Chemical shifts are reported in ppm relative to standards (TMS for samples in CDCl$_3$ and [$^2$H]$_7$-DMF; TMS or DMSO for samples in [$^2$H]$_6$-DMSO; external 85% aqueous phosphoric acid for $^{31}$P spectra; TSP or HDO for samples in D$_2$O). $^{31}$P NMR spectra were recorded on JEOL FX-90Q or EX-400 spectrometers. Multiplicities are

indicated as follows: s (singlet), d (doublet), t (triplet), dd (doublet of doublets), q (quartet), m (multiplet), td (triplet of doublets), tt (triplet of triplets) and br (broad signal). Routinely, NMR resonances were formally assigned by use of multi-dimensional techniques and for hydroxyl protons through $D_2O$ exchange. Melting points were determined using a Reichert-Jung Thermo Galen Kofler block and are uncorrected. Elemental analysis was carried out at the University of Bath microanalysis service. Mass spectra were obtained by fast atom bombardment (FAB) in a VG707E spectrometer. Ion-exchange chromatography was carried out on a LKB-Pharmacia medium pressure ion-exchange chromatograph on Q Sepharose Fast Flow resin eluting with a gradient of triethylammonium bicarbonate (TEAB) buffer. Fractions containing target compound were detected and quantified by a modification of the Briggs phosphate assays[57]. All final target compounds studied were used as their triethylammonium salts and were of >95% purity as judged by HPLC methodology as described[58]. Briefly, compounds were detected by post-column complexation with ferric ion after anion-exchange separation on a CarboPac PA200 (Dionex) column eluted with methane sulfonic acid[59].

### Scyllo-inositol 1,2,3,5-tetrakisphosphate (1-p)

**2,4-Di-O-p-methoxybenzyl-1,3,5-O-methylidyne scyllo-inositol (13).** Ketone **12**[37] (1.98 g, 4.63 mmol) was dissolved in a mixture of THF (20 mL) and methanol (80 mL). Sodium borohydride (430 mg, 11.6 mmol) was added gradually and the mixture stirred at r.t. for a further 30 min. TLC showed complete conversion into a product with $R_f$ 0.44 (ethyl acetate/hexane 1:1). Water (100 mL) was added and the product extracted with dichloromethane (3 × 100 mL). The combined organic phases were washed with brine and dried over $MgSO_4$. Evaporation of solvents under reduced pressure gave a white solid, which was recrystallized from ethyl acetate/hexane to provide **13** (1.77 g, 4.12 mmol, 89%).

**1,3-Di-O-p-methoxybenzyl-scyllo-inositol (14).** To a solution of **13** (1.0 g, 2.32 mmol) in methanol (50 mL) was added 1 M HCl (5 mL). The mixture was heated at reflux for 30 min after which TLC showed that most of the starting material ($R_f$ 0.48, ethyl acetate/ hexane 1:1) had been consumed. The heating source was removed and concentrated ammonia solution (1 mL) was added. Stirring was continued for a further 30 min at r.t. and then the solvents were removed by evaporation under reduced pressure to give a solid residue which was extracted with hot ethyl acetate (5 × 50 mL). Evaporation of the combined extracts, followed by crystallization from methanol/ethyl acetate gave **14** (666 mg, 1.58 mmol, 68%).

**4,6-Di-O-p-methoxybenzyl-scyllo-inositol 1,2,3,5-tetrakis[bis(2-cyanoethyl)phosphate] (15).** To a solution of bis(cyanoethoxy)diisopropylaminophosphine (516 mg, 1.90 mmol) in dry dichloromethane (2 mL) was added 1H-tetrazole (267 mg, 3.80 mmol). The mixture was stirred at room temperature for 10 min and then the tetrol **14** (100 mg, 0.238 mmol) was added. The mixture was stirred for a further 1 h, after which a $^{31}P$ NMR spectrum showed a complex pattern of signals around 141 ppm. The mixture was cooled to −78 °C and m-CPBA (360 mg, 2.09 mmol) was added. The mixture was allowed to reach room temperature, and then diluted with ethyl acetate (50 mL). The solution was washed with 10% sodium sulphite solution, sat. $NaHCO_3$ and brine (50 mL of each), dried ($MgSO_4$) and evaporated in vacuo to give an oil. Purification by column chromatography (ethyl acetate/ethanol 5:1) gave the tetrakisphosphate triester **15** as a colorless oil (237 mg, 0.202 mmol, 85%).

**Scyllo-inositol 1,2,3,5-tetrakisphosphate (1-p).** Using a three-necked flask, ammonia (~100 mL) was condensed at −78 °C, and a sodium excess was added for drying. The resulting deep blue solution was stirred for 30 min at −78 °C. Using a second three-necked flask kept at −78 °C a small volume (~30 mL) of the dry ammonia was then distilled over and sodium added, ensuring the solution remained blue-black for 10 min. Compound **15** (100 mg, 85.8 μmol), dissolved in dry dioxane (2 mL), was then added to the vigorously stirred mixture and after 5 min, the reaction quenched with methanol and then deionized water, whereupon solvents and ammonia were removed in vacuo. The residue, dissolved in deionized water (300 mL), was purified by ion-exchange chromatography on Q Sepharose Fast Flow resin, eluting with a triethylammonium bicarbonate buffer gradient (0 to 1 moldm$^{-3}$) at pH 8.0. The target **1-p**, as its glassy triethylammonium salt obtained after evaporation in vacuo, was eluted between 670 mM and 780 mM. Yield 61 μmol, 71%.

### DL-6-Deoxy-6-hydroxymethyl-scyllo-inositol 1,2,4-trisphosphate (racemic 5-s)

**DL-6-Deoxy-1,3-di-O-benzyl-6-benzyloxymethyl-2,4-di-O-p-methoxybenzyl-scyllo-inositol (17).** To an ice-cold mixture of compound **16** (300 mg, 0.427 mmol) and 4 Å molecular sieves (2 g) and THF (10 mL) under $N_2$ were added borane-trimethylamine complex (190 mg, 2.60 mmol) and freshly pulverised aluminium chloride (345 mg, 2.59 mmol). The mixture was stirred at 0 °C for 23 h, after which time TLC (ethyl acetate/hexane 1:1) showed the reaction to be complete with conversion of starting material ($R_f$ 0.57) into a product ($R_f$ 0.49). Ether (50 mL) was added followed by ice-water (50 mL) and 1 M HCl (10 mL). The organic layer was removed and the aqueous layer re-extracted with a further 50 mL of ether. The combined organic extracts were washed with brine (100 mL), dried ($MgSO_4$) and evaporated under reduced pressure to give an oily residue. Purification by flash chromatography (dichloromethane/ethyl acetate 20:1) gave the alcohol **17** as a colorless oil, which slowly solidified (195 mg, 0.271 mmol, 65%).

**DL-6-Deoxy-1,3-di-O-benzyl-6-benzyloxymethyl-scyllo-inositol (18).** The alcohol **17** (200 mg, 0.284 mmol) was dissolved in ethanol (60 mL) and 1 M HCl (30 mL) added. The mixture was heated at reflux for 5 h and then the solvents removed by evaporation under reduced pressure. The residue was dissolved in dichloromethane (50 mL), washed with sat $NaHCO_3$ and brine (50 mL of each) and evaporated to give an oily residue. Purification by flash chromatography (ethyl acetate/hexane 1:1) gave triol **18** as a white solid (115 mg, 0.248 mmol, 87%).

**DL-6-Deoxy−3,5-di-O-benzyl-6-benzyloxymethyl-scyllo-inositol 1,2,4-tris(dibenzylphosphate) (19).** To a solution of bis(benzyloxy) diisopropylaminophosphine (356 mg, 1.03 mmol) in dry dichloromethane (3 mL) was added 1H-tetrazole (144 mg, 2.06 mmol). The mixture was stirred at room temperature for 20 min and then the triol **18** (80 mg, 0.172 mmol) was added and stirring was continued for 30 min. The mixture was cooled to −78 °C, m-CPBA (200 mg, 1.16 mmol) was added, and the cooling bath was removed. The mixture was allowed to reach rt and then diluted with ethyl acetate (50 mL). The clear solution was washed with 10% $Na_2SO_3$, 1 M HCl, sat. $NaHCO_3$ and brine (50 mL of each) dried ($MgSO_4$) and evaporated in vacuo, giving an oily residue. Purification by column chromatography (chloroform acetone 10:1) afforded **19** (183 mg, 0.147 mmol, 85%) as a colorless oil which slowly crystallized.

**DL-6-Deoxy-6-hydroxymethyl-scyllo-inositol 1,2,4-trisphosphate (racemic 5-s).** The trisphosphate triester **19** (60 mg, 48 μmol) was deprotected as described for compound **1-p**. Purification by ion-exchange chromatography on Q Sepharose Fast Flow Resin, as before gave the glassy triethylammonium salt of **5-s**, which eluted between 450 mM and 550 mM TEAB. Yield 34 μmol, 71%.

**DL-6-Deoxy-6-phosphoryloxymethyl-*scyllo*-inositol 1,2,4-tri-sphosphate (racemic 5-p)**

**DL-6-Deoxy-6-hydroxymethyl-1,3-di-*O*-benzyl-*scyllo*-inositol (20).** To **16** (400 mg, 0.57 mmol) in EtOH (20 mL) was added 1 M hydrochloric acid (10 mL). [NB for the full associated synthetic details to **16** from *myo*-inositol orthoformate for this route please see Supplementary Methods and Supplementary Fig. 2]. The mixture was heated to reflux for 3 h. Evaporation then chromatography ($CHCl_3$/MeOH 9:1) gave the tetrol **20** (152 mg, 71%) as a white solid.

**DL-6-deoxy-6-(dibenzyloxyphosphoryloxy)methyl-3,5-di-*O*-benzyl-*scyllo*-inositol 1,2,4-tris(dibenzylphosphate) (21).** To a solution of bis(benzyloxy)diisopropylaminophosphine (1.11 g, 3.21 mmol) in dry $CH_2Cl_2$ (3 mL) was added 1*H*-tetrazole (450 mg, 6.42 mmol). The mixture was stirred at room temperature for 30 min, then the tetrol **20** (150 mg, 0.401 mmol) was added. After 40 min the mixture was cooled to −78 °C and *m*-CPBA (57-86%, 1.1 g) was added. The mixture was allowed to reach room temperature and stirring was continued for 30 min, then $CH_2Cl_2$ (50 mL) was added. The solution was washed with 10% $Na_2SO_3$ solution (50 mL), saturated $NaHCO_3$ solution (50 mL) and brine (50 mL) then dried with $MgSO_4$. Evaporation followed by flash chromatography ($CHCl_3$/acetone 5:1) gave **21** (410 mg, 72%) as a colorless oil.

**DL-6-Deoxy-6-phosphoryloxymethyl-*scyllo*-inositol 1,2,4-trisphosphate (racemic 5-p).** To **21** (150 mg) in MeOH (40 mL) and water (10 mL) was added 10% palladium on activated charcoal (200 mg). The mixture was shaken under hydrogen at 40 psi at room temperature overnight. The mixture was then filtered through Celite and the solvents were evaporated. The residue was dissolved in water (100 mL) and purified by ion-exchange chromatography on Q Sepharose Fast Flow resin eluting with a gradient of triethylammonium hydrogen carbonate buffer (0–1 moldm$^{-3}$). Evaporation gave the tetrakisphosphate **5-p** (79 $\mu$mol, 75%) as a colorless glass.

The following other ligands investigated were synthesized as described, purified by ion exchange chromatography, quantified using a Briggs phosphate assay and used as their triethylammonium salts; they were >95% pure by NMR spectroscopy: L-*scyllo*-inositol 1,2,4-trisphosphate **1-s**[60], D-*myo*-inositol 1,4,6-trisphosphate **2-s**[34], D-3-deoxy-*myo*-inositol 1,4,6-trisphosphate **3-s**[36], L-*chiro*-inositol 2,3,5-trisphosphate **4-s**[35], β-D-glucopyranosyl 1,3,4-trisphosphate **6a-s**[59], α-D-glucopyranosyl 1,3,4-trisphosphate **6b-s**[59], β-D-glucopyranosylmethanol 3,4,1′-trisphosphate **7a-s**, α-D-glucopyranosylmethanol 3,4,1′-trisphosphate **7b-s**[44,45], ribophostin **8** and D-*chiro*-inositol ribophostin **9**[28].

## Purification of IP3K samples

A construct encoding human IP3K-KD A isoform (residues 187–461) into pOPTG vector was used[16]. The protein was expressed in *Escherichia coli* BL21 star (DE3) strain. Cells were grown at 37 °C to OD600 = 1.0 then induced with 0.4 mM IPTG for 18 h at 16 °C. Protein purification was performed by adapting the reported protocol[16]. Cells were resuspended in buffer A (10 mM Tris pH 7.5 [4 °C], 200 mM NaCl and 2 mM DTT) and disrupted by sonication. Clarified cell lysate was diluted 1/4 in buffer B (10 mM Tris pH 7.5 [4 °C] and 2 mM DTT) and loaded into a 5 mL HiTrap heparin column (GE Healthcare) washed by buffer C (20 mM Tris pH 7.5, 50 mM NaCl and 2 mM DTT), and eluted with a 0.05 M–1 M salt gradient. The eluted sample was subsequently loaded on to a 5 mL glutathione–Sepharose HP column (GE Healthcare), washed with buffer A and eluted with reduced glutathione 10 mM (Sigma-Aldrich) in buffer A. The GST tag was cleaved with home-made TEV (tobacco etch virus) protease overnight at 4 °C, (1:40 TEV/protein mass ratio). After glutathione removal with a PD10 column, the cleaved protein was further purified using a 5 mL glutathione–Sepharose HP column (GE Healthcare) followed by gel filtration on a 16/600 Superdex200 (GE Healthcare) column

equilibrated in buffer D (20 mM Tris pH 7.5 [4 °C], 0.05 M $(NH_4)_2SO_4$ and 2 mM DTT). Protein samples were analyzed by SDS/PAGE, concentrated around 18 mg/mL and stored at −80 °C until use.

## Crystallization of IP3K complexes and structural determination

We produced well-diffracting IP3KA-KD crystals starting from the reported crystallization conditions[16]. Optimal crystals were obtained at 18 °C using the vapor diffusion method in sitting drop from 48-well plates (Hampton) and as precipitant solution 0.80–0.84 M tri-sodium citrate, 0.1 M Tris-HCl pH 8.5 (r.t.) and 0.1 M NaCl. To prepare ligand-protein complexes, we gradually changed the mother liquor of the crystal drops to the soaking solution (1.8 M $Li_2SO_4$ and 100 mM Tris-HCl pH 8). The last addition of soaking solution contained the ligands: nucleotide (ATP, AMPPNP or ADP depending on the complex, all of them from Sigma), inositide (Fig. 1 and Supplementary Fig. 1) and divalent ion (from $MgCl_2$ or $MnCl_2$ depending on the complex). Crystals were left, from 2 h to o/n, in the soaking solution. The soaking solution was itself a cryoprotectant. For a standard experiment we used 3 mM nucleotide, 3 mM divalent ion and 5 mM $InsP_3$ analogue. Standard experiments were successful with compounds **1**–**4** series, **6a**–**s**, **6b**–**s** and **7a**–**s** (Fig. 1). When using ligands that were not well captured in previous conditions because of poor binding or their status as racemates, as in the case of compounds **5-s** and **5-p**, we used 3 mM nucleotide, 3 mM divalent ion and 10 mM inositide (**5-s**) or 30 mM inositide in the citrate condition using 30% glycerol as cryoprotectant (**5-p**). Finally, conditions to produce reactions in the crystallization experiments from atypical substrates (**6a**–**s** and **7a**–**s**) to get the products (**6a**–**p** and **7a**–**p**) in the crystals were: 10 mM ATP, 10 mM divalent ion and 7 mM inositide into the soaking solution. Complete datasets from crystals were collected at 100 K in ALBA and ESRF synchrotron facilities. Datasets were processed using XDS[61] and scaled with Aimless from the CCP4 suite[62] or with autoPROC[63]. The crystal complexes structures were solved using Synthesis Fourier Difference using the coordinates of IP3KA-KD from PDB database (PDB code 1w2c). Several rounds of refinement using Refmac5[64] were alternated with model building using COOT[65]. The final processing and refinement statistics for best datasets obtained are reported in Supplementary Table 1. The crystals show two molecules in the asymmetric unit, in all cases molecule A being much more ordered than molecule B. Therefore, all our conclusions are based on the analysis of molecule A of the complexes in which electron density maps allowed unambiguous fitting of all the ligands. When the ligand structure was not in databases, ligand coordinates and cif files were obtained using PyMOL, COOT and Mercury programs[65–67] and Grade server (https://grade.globalphasing.org/cgi-bin/grade2_server.cgi). The images were prepared with PyMOL.

## Fluorescence thermal shift analysis

Thermal protein denaturation was documented by recording intrinsic protein fluorescence at 330 and 350 nm wavelengths. The fluorescence recorded in each thermal run is plotted as 350/330 nm ratio and used to calculate the *Inflection Temperature* ($T_i$). Ti is the temperature at which a protein undergoes a transition in its folding state. This calculation is performed automatically by the Tycho NT.6 software. The protocol optimized for this procedure, involves a heating time of 2 min, a sample volume of 10 µl placed in a very thin capillary, and a temperature ramp from 35 °C (308 K) to 95 °C (368 K). For the experiment, IP3K-KD was diluted with buffer D to achieve a concentration of 0.31 mg/mL (10 µM). It was then mixed with 100 µM of chiral ligands or 200 µM of racemic ligands (all synthesized as stated in this work except $InsP_3$ and $InsP_4$ that were commercial from Avanti Polar Lipids), AMP-PNP (substrate analogues) or ADP (product-like ligands) and 100 µM $MgCl_2$. Microsoft Excel was used to calculate and plot $\Delta T_i$ values and standard deviations.

## HPLC analysis

For HPLC experiments, 150 nM IP3K-KD was incubated with 500 μM inositol phosphate substrate or analogue and 1 mM ATP under ATP-regenerating conditions in 20 mM HEPES pH 7.5, 1 mM MgCl$_2$ for 2 h at 30 °C, with subsequent HPLC analysis as described[58].

## Ligand displacement assays

Binding of 2-FITC-InsP$_3$, a small molecule, to a much larger protein slows the 'tumbling' of the probe on timescale comparable to the lifetime of its fluorescence. This results in an increase in the polarization (or anisotropy) of fluorescence emission. Binding of a competing/displacing ligand to protein is observed as a decrease in the polarization (anisotropy) of 2-FITC-InsP$_3$. A comparison of the strength of binding of enzyme substrates, analogues and products to IP3K-KD was obtained by measurement of fluorescence polarization (here anisotropy) of 2-FITC-InsP$_3$/IP3K-KD complex. Experimentally, 100 nM IP3K-KD was mixed with 2 nM 2-FITC-InsP$_3$[37] in 20 mM HEPES pH 7.5, 1 mM EDTA and 50 mM KCl with the addition of displacing compounds over a range of 1 nM–50 μM. Assays were analyzed using a ClarioSTAR plate reader and IC$_{50}$ values determined from fit of the data to a 4-parameter logistic as described[68]. From IC$_{50}$ values, we have estimated the K$_i$ (K$_d$) values for the displacing ligands using the equation of Cheng and Prusoff:[69]

$$K_i = IC_{50} / \left(1 + \frac{[L]}{K_d}\right) \tag{1}$$

where [L] is the ligand (2-FITC-InsP$_3$) concentration and K$_d$ the dissociation constant for its interaction with IP3K-KD. Fitting of the binding of 2-FITC-InsP$_3$ to IP3K-KD with a one-site binding model in GraphPad Prism v.6 (GraphPad Software Inc., San Diego, USA) yielded a K$_d$ of 68 nM. Consequently, the K$_i$ values for the displacing ligands differ trivially from their IC$_{50}$ values.

## Docking

Docking experiments and visualization were performed with *GOLD* and *Hermes*, programs respectively (CCDC Software Ltd[70].). GOLD (*Genetic Optimisation for Ligand Docking*) uses a genetic algorithm to explore a wide range of ligand conformations able to interact with the target protein active site. GOLD uses a quality function of fit that allows a solution score considering the bonds generated in the docking[70]. We chose the default *ChemPLP* function because it has optimized the time consuming and success degree ratio for prediction of ligand poses. The *ChemPLP* function score is dimensionless and takes into account the number and geometry of hydrogen bonds, hydrophobic contact area, ligand conformation and interactions with metal or water molecules to evaluate ligand protein fitting[70]. To make a comparison of the different ligands, the *ChemPLP* function scores were normalized against the molecular weight of each ligand. For our analysis, we added hydrogen atoms to the ligands, defined all protein conformations as rigid bodies and ligands as flexible elements (except ring torsions that have been kept rigid), and kept the nucleotide and Mn$^{2+}$/Mg$^{2+}$ atoms in the active site as intrinsic constituents. First, we made an initial validation of GOLD in our system, checking the reproducibility of experimental complexes structures obtained in this work. We did this in two ways: (1) using each of the ligands with the protein conformation in their complexes. Regarding water molecules, we input those bridging protein and ligand, allowing rotation freedom and intermittence to keep them or not. The active site was dimensioned to 6 Å around the ligand position and we searched for 10 solutions in each ligand run. The rest of the parameters were used as program default. (2) An ensemble docking that tests all ligands against all protein conformations obtained in experimental structures. The active site was defined as all atoms in a 10 Å radius sphere centered on the inositol ring (centroid

x = 12.17 Å, y = 28.52 Å, z = 84,28 Å). We selected eight water molecules with intermittence possibility, rotation freedom and translation to 1.5 Å. We searched for 50 solutions for each ligand, keeping the remaining parameters as before. We have repeated the procedure with the latter settings searching the ligand poses against the protein structure of the complex IP3K:InsP$_3$ (1w2c) in order to compare with the ensemble docking. We checked that this method is suitable for predicting the main characteristics of IP3K-KD substrate analogues interactions (Supplementary Fig. 6, 7), but found it was unsuitable to predict product and product analogues interactions. Finally, we built other putative IP3K ligands coordinates and cif files using Mercury and Grade Server as described before. We performed docking experiments in a similar way to ensemble docking, considering in this case the 10 best solutions.

## Data availability

The authors declare that the main data supporting the findings of this study are available within the article and its Supplementary Information files. The atomic coordinates and structure factors of all structures generated in this study have been deposited in the PDB database (Research Collaboratory for Structural Bioinformatics, Rutgers University, New Brunswick, NJ (www.rcsb.org/)) under accession codes 8PP8, 8PP9, 8PPA, 8PPB, 8PPC, 8PPD, 8PPE, 8PPF, 8PPG, 8PPH, 8PPI and 8PPJ. The atomic coordinates used in this study are available in the PDB database under accession codes 1w2c, 1w2d, 1w2f and 5w2i. Source data are provided with this paper.

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

## Acknowledgements

We would like to acknowledge Paula Sanz Benito for valuable technical assistance. We are grateful to the staff of the Synchrotron Radiation Sources at Alba (Barcelona, Spain) for providing access and for technical assistance at BL13-XALOC beamline. We also acknowledge the European Synchrotron Radiation Facility ESRF (Grenoble, France) and the staff of beamlines ID23-1 and ID23-2 for assistance and support. M.A.M.-M. and this work has been supported by grants PID2020-117400GB-100 and BFU2017-89913-P from the Spanish Ministerio de Ciencia e Innovación and Ministerio de Economía y Competitividad. B.V.L.P is a Wellcome Trust Senior Investigator (grant 101010). This research was supported by The Wellcome Trust. For the purpose of Open Access, the authors have applied a CC BY public copyright license to any Author Accepted Manuscript version arising from this submission.

## Author contributions

B.G., C.A.B. and B.V.L.P. designed and supervised the research. M.A.M.-M. performed the constructs preparation, protein purification, crystallization, data collection, structural determination, docking analysis and thermal shift experiments. R.O.-G. helped in the constructs and sample preparation, protein crystallization and data collection. L.I. supervised docking analysis. S.G., A.M.R. & M.L.S. synthesized the inositol and carbohydrate compounds with supervision from B.V.L.P. and H.W. performed the HPLC and fluorescence polarization assays with supervision of C.A.B. B.G. and M.A.M.-M. analyzed the crystallographic data. M.A.M.-M., B.V.L.P. and B.G. wrote the paper. M.A.M.-M., L.I., M.L.S., C.A.B., A.M.R., B.V.L.P. and B.G. edited the manuscript.

## Competing interests

The authors declare no competing interests.
