## [Peer Review File · Nature Communications]

REVIEWER COMMENTS

Reviewer #1 (Remarks to the Author):

The manuscript by Márquez-Moñino et al. presents unprecedented insight into the recognition of substrates and products for the enzyme IP3KA. These insights help understand the enzymatic mechanism of IP3KA, and they will be useful for design and selection of specific inhibitors and activators of the enzyme. The authors have synthesised a broad panel of substrate analogues to probe the specificity and plasticity of the IP3 binding site of IP3KA. This tremendously augments what is known about this enzyme that has been shown to be vital for neuronal function. This panel was carefully designed to test the ability of the enzyme to accommodate substituents at all the IP3 positions, and the impressive synthetic routes are clearly described in the manuscript. What gives the manuscript unique insight is that the authors have been able to determine enzyme/ligand structures for almost all the compounds, and they have used this structural information to try to predict the binding for an even broader group of ligands. The authors have clearly gleaned from their study how it would be possible to make ligands that would differentiate between IP3-receptor and IP3KA binding. They have shown that the enzyme can accommodate a surprising range of substituents at the 3-position of the inositol. Nevertheless, the IP3KA active site shows only limited structural variation, and it binds its ligands in only one way, making it distinct from the much more widely studied inositol polyphosphate multikinases. A particularly interesting observation made by the authors is that pyranose-based derivatives are also good ligands for IP3K. This suggests that ligands can be synthesized generally in chiral form and protecting group chemistry is usually much easier than working with inositol derivatives. Overall, their structural study revealed a moderate IP3K plasticity enabling them to identify sites to which unusual IP3K ligands could be targeted that could even yield unexpected IP3K products.

The manuscript is clearly written, and the authors have done an admirable job throughout the text to avoid confusing casual readers with the wide variation in numbering that arises from the systematic nomenclature. Within the main text, the authors have stuck with IP3-based numbering. The text is well written, and I would suggest only a few minor changes as follows.

1. Lines 31-32 of the abstract. The authors state that: "modified ligands exploit active site plasticity to generate a 'spring effect'." I am not sure that this choice of wording evokes any real understanding of the conformational changes. In at least one place in the text, I think it actually tends toward generating misunderstanding (see below).
2. Line 81. "Despite all this it is..." Needs a comma after "this"
3. Line 164. "...pseudo 1,2,3,4,5 or 6 positions which..." Should have a comma before "which"
4. Line 180 the "spring effect" has been introduced. However, there is no explanation as what the authors mean by this.

From the Fig. 4a, it looks like helix alpha 2I is moving. However, a naïve reader would likely see a helix and think the helix is a spring that stretches. What Fig 4a seems to show is that entire helix shifts in a direction parallel to its axis. This could be a spring-like motion, but a spring that would produce such a motion would be in the linkers that connect the helix to the rest of the IP insertion. The authors need to state precisely what they mean by "spring effect".

5. Line 192 seems a bit awkward.

"could explain the fitting of compounds in the InsP3-site or another (Fig. 3)."

Might be better as "could account for variations in the compound binding relative to InsP3."

6. Lines 196-197 "Some of the compounds analyzed are in fact product mimics such as 1-p, 5-p, 6a-p and 7a-p (Fig. 1 and 3)."

It would be better to state: "Some of the compounds analyzed, such as 1-p, 5-p, 6a-p and 7a-p, are in fact mimics of the natural product InsP4 (Fig. 1 and 3)."

7. Lines 198-200: "Whereas complexes with the former two were obtained by crystal soaking with the product analogue, the latter two were obtained when crystals were soaked into solutions containing the substrate mimics 6a-s and 7a-s under specific conditions (see below)."

Might be better as: "The complexes with 1-p and 5-p were obtained by soaking crystals directly with these product analogues. In contrast, complexes with 6a-p and 7a-p were obtained when crystals were soaked into solutions containing the substrate mimics 6a-s and 7a-s under specific conditions (see below)."

8. Figure 4a legend: "(Right) A picture of the IP-lobe from the IP3K complexes with 6a-p and 5-p showing the maximum variation experimented by the a2I from the IP-lobe, an effect named in this work as a spring effect." Is awkward. A better version might be:

"(Right) A picture of the IP-lobe from the IP3K complexes with 6a-p and 5-p, showing the maximum shift of the IP-lobe helix a2I. This shift is the "spring effect" referred to in the text."

9. Figure 5. It would be helpful to have Kds for the ligands based on the Cheng and Prusoff approximation, in addition to the IC50s.

10. Line 257 "Analysis reveal..." should be "Analysis revealed..."

11. Line 281 "...were examined as protein substrates" should be "were examined as IP3K substrates".

12. Line 346 "...causing a shift into the IP3K active site..." would be better as "...causing a shift in the IP3K active site..."

13. Legend for Ext. Fig. 3., line 1229: "...stabilizing the bi-substrate binding...". I think that the usage of the term "bi-substrate" in this figure is misleading, and it would be better to rephrase it. Here, bi-substrate refers to ATP and 3-s ligand; on line 396, I think "bi-substrate" refers to covalent ANP-ligand compounds.

14. Legend for Extended Fig 4a. Says: "a, GOLD best 10 docking solutions (green/yellow/red lines) for each ligand using its corresponding experimental IP3K-KD structure."

This seems to say that GOLD was predicting binding based on the experimental structure for a given ligand (for each ligand the authors removed the ligand from the experimental structure and ran GOLD to "predict" how the ligand binds). What were the results if GOLD were given just the InsP3(InsP4)/IP3KA structures with the ligands removed as the target?

In the methods for docking, it says that docking was done in two ways:

1. Using the observed conformation of the enzyme corresponding to the particular ligand;
2. Using an ensemble of all the conformations observed for the enzyme.

However, the legend for Extended Fig. 4 seems to refer only to the first method. What were the results of the second method?

15. Legend for the Ext Fig. 4b. The authors state:

"GOLD best solutions for compounds 7b-s, 8 and 9 bound in IP3K active site that show how phosphates mimic InsP3 phosphate positions. In these three cases the conformations of the ligands present non-favored torsions explaining why we have not detected significant binding in any of our experiments and they failed to form a crystal complex."

Which torsions are the authors referring to? Are these the chair/boat/twist-boat conformations of the ligand? Given the overall observation that GOLD was not able to dock product ligands confidently, even though it was given the conformation of the enzyme from the enzyme/product complex, it does not seem as though the docking poise predicted by GOLD for 7b-s, 8, 9, 10, and 11 should be taken too seriously.

It seems as though the authors regard failure to get a ligand/protein structure by crystallography as an indication that a compound is not a substrate. Fig. 6 shows a product peak for 7b-s in the HPLC

analysis, suggesting that it is acting as a substrate. It is a reasonably common experience for even ligands that are good substrates of enzymes not to bind to an enzyme in crystals, due to crystallisation reagents or crystal packing.

14. Line 472. The authors refer to "..., either primary or secondary position..." This would be better as "..., either as a primary or secondary hydroxyl, ..."

Reviewer #2 (Remarks to the Author):

This study takes advantage of the complementary expertise of three research groups in inositol phosphate biochemistry, organic synthesis, and structural biology. The authors characterised the ability of the IP3K to accommodate and to metabolise a large array of IP3/4-mimics using multiple approaches. Surprisingly, this investigation revealed an unexpected IP3K enzymatic plasticity. The ability of IP3K to phosphorylate 'pseudo' 3-position even on non-ciclytol carbohydrate substrate is of primary importance. The paper follows a clear experimental logic and is overall easy to read for the specialist. Passages where the chemical nomenclature joins the structural data are understandable only thanks to the good quality of the figures otherwise it would be difficult even for the specialist. However, given the technical nature of the arguments discussed, I do not think that the text could be simplified. I did not find fault in the presented experiments that adequately addressed the paper's questions. Nevertheless, I list below a few issues that require the author's attention.

My main concern is the lack of any comparisons with other IP-kinase such IPMK and ITPKs. To better appreciate the newly discovered IP3K enzymatic plasticity the introduction should compare IP3K substrate selectivity with the substrate 'promiscuity' of IPMK/ITPKs. Similarly, the discussion should comment (or speculate) on what the authors discovered for IP3K means for IPMK/ITPKs enzymology.

Line 57 please define CAM. It is +/- defined on line 61 where instead should appear as CAM.

Lines 105-114 The authors should try to simplify the very long note in [bracket].

Line 188 I would be more specific and define an equatorial 2OH or pseudo-2OH substituent.

Line 293 While the unexpected discovery that -InsP3 analogues substituted at the pseudo 3-position- is quite important, it does not need to be written in bold. It did not represent the title of a new paragraph.

Figure 1 and extended Fig.1 I suggest boxing the bold -Natural substrate and Natural product- labels, to be consistent with the other boxed bold labels. Additionally, I would define also in the figure legend (not just as a figure's element) the meaning of the colored yellow, red, and green arrows. Only the green arrow is defined.

Reviewer #3 (Remarks to the Author):

This is a very ambitious paper that pushes our understanding of ligand binding to inositol 1,4,5-trisphosphate kinase. Extended Fig 1 shows the analogs of the natural substrate that have been prepared. These syntheses of these compounds is non-trivial, but worth the effort. Once in hand, Fig 5 and Fig 6 show how these compounds interact with their target in the biochemical assays that quantify ligand binding. Crystallography and docking follow that support a structural model for all of the data, culminating in a "spring effect" that I think the authors mean to correlate with the qualitative plasticity of the active site - that is, in order to accommodate the required criteria for good binding (enumerated in the manuscript and conclusion), the enzyme undergoes some conformational changes to accommodate the unusual substrates. A highlight of the paper is the extensive presentation of substrate superpositions to show similarities and differences (for both substrate and enzyme) upon binding.

I think this is a superb contribution and I recommend publication in Nat Comm essentially as is. The paper is particularly welcome in the ambition of the synthetic chemistry, which for this study involves multi-step synthesis for each compound. Too few studies of this type appear nowadays, as the chemistry investment is considered too high for most groups. Without this work, the most unique observation in the paper, that a methylene (CH₂)-extended substrate is still competent, would not have been unearthed.

If I were to request an addition/revision to the paper, I think some of the forward-looking statements about other substrates that could reveal new biochemistry could be less vague. What do these new results make the authors curious about? My guess is that they have lots of ideas, but maybe they could speculate more on what the specific results of this paper presage.

Reviewer #1:

We greatly appreciate the comments from reviewer #1, and provide our responses below that we trust will deal with his/her comments:

1. Lines 31-32 of the abstract. The authors state that: “modified ligands exploit active site plasticity to generate a ‘spring effect’.” I am not sure that this choice of wording evokes any real understanding of the conformational changes. In at least one place in the text, I think it actually tends toward generating misunderstanding (see below).

Yes, we agree with this observation and have replaced the term “spring effect” with “helix-tilt” in the abstract. We have introduced an explanation for the “spring-effect” in the Results section (lines 181-183). Please see our answer to point 4 below for a more detailed explanation.

2. Line 81. “Despite all this it is...” Needs a comma after “this”.

Corrected (line 88)

3. Line 164. “...pseudo 1,2,3,4,5 or 6 positions which...” Should have a comma before “which”.

Correction made (line 166)

4. Line 180 the “spring effect” has been introduced. However, there is no explanation as what the authors mean by this.

From the Fig. 4a, it looks like helix alpha 2I is moving. However, a naive reader would likely see a helix and think the helix is a spring that stretches. What Fig 4a seems to show is that entire helix shifts in a direction parallel to its axis. This could be a spring-like motion, but a spring that would produce such a motion would be in the linkers that connect the helix to the rest of the IP insertion. The authors need to state precisely what they mean by “spring effect”.

The reviewer is right, we appreciate this observation. As the reviewer points out, the term “spring effect” should be clearly described to avoid confusion. In the abstract, we have changed “generate a spring-effect” with “generating a helix-tilt”. Indeed, the parts acting as a spring are the hinges that connect the $\alpha 2I$ helix to the protein structure, resulting in a helix tilt of 9 degrees. We have included a detailed description of the “spring effect” in the manuscript (lines 181-183) and added a new image in Figure 4a, as well as a supporting video (Supplementary Movie 1) in the Supplementary files, to properly illustrate this conformational change.

5. Line 192 seems a bit awkward. “could explain the fitting of compounds in the InsP3-site or another (Fig. 3).” Might be better as “could account for variations in the compound binding relative to InsP3.”

Corrected in accordance with the reviewer’s suggestion (line 195)

6. Lines 196-197 “Some of the compounds analyzed are in fact product mimics such as 1-p, 5-p, 6a-p and 7a-p (Fig. 1 and 3).” It would be better to state: “Some of the compounds analyzed, such as 1-p, 5-p, 6a-p and 7a-p, are in fact mimics of the natural product InsP4 (Fig. 1 and 3).”

Corrected in accordance with the reviewer’s suggestion (lines 199-200)

7. Lines 198-200: “Whereas complexes with the former two were obtained by crystal soaking with the product analogue, the latter two were obtained when crystals were soaked into solutions containing the substrate mimics 6a-s and 7a-s under specific conditions (see below).”

Might be better as: “The complexes with 1-p and 5-p were obtained by soaking crystals directly with these product analogues. In contrast, complexes with 6a-p and 7a-p were obtained when crystals were soaked into solutions containing the substrate mimics 6a-s and 7a-s under specific conditions (see below).”

Corrected in accordance with the reviewer’s suggestion (lines 201-203)

8. Figure 4a legend: “(Right) A picture of the IP-lobe from the IP3K complexes with 6a-p and 5-p showing the maximum variation experimented by the $\alpha 2I$ from the IP-lobe, an effect named in this work as a spring effect.” *Is awkward.*

A better version might be: “(Right) A picture of the IP-lobe from the IP3K complexes with 6a-p and 5-p, showing the maximum shift of the IP-lobe helix $\alpha 2I$. This shift is the “spring effect” referred to in the text.”

Corrected in accordance with the reviewer’s suggestion (line 1191)

9. Figure 5. *It would be helpful to have Kds for the ligands based on the Cheng and Prusoff approximation, in addition to the IC50s.*

We have estimated the K_i (K_d) from IC_{50} values using the suggested approximation and estimated the confidence interval through simple linear approximation. The K_d s obtained are very similar to IC_{50} values, in this case. We have detailed this procedure in the Methods section (lines 815-822), included the K_d values in panel c of Figure 5, and presented the values along with the confidence intervals in Supplementary Table 2.

10. Line 263 “Analysis reveal...” *should be “Analysis revealed...”*

Corrected (line 261)

11. Line 281 “...were examined as protein substrates” *should be “were examined as IP3K substrates”.*

Corrected (line 285)

12. Line 346 “...causing a shift into the IP3K active site...” *would be better as “...causing a shift in the IP3K active site...”*

Corrected (line 361)

13. Legend for Ext. Fig. 3., line 1229: “...stabilizing the bi-substrate binding...”. *I think that the usage of the term “bi-substrate” in this figure is misleading, and it would be better to rephrase it. Here, bi-substrate refers to ATP and 3-s ligand; on line 396, I think “bi-substrate” refers to covalent ANP-ligand compounds.*

Done, changed by “stabilizing both, ATP and InsP analogue” (line 255 of the Supplemental Files, in Supplementary Fig. 4)

14. Legend for Extended Fig 4a. *Says: “a, GOLD best 10 docking solutions (green/yellow/red lines) for each ligand using its corresponding experimental IP3K-KD structure.” This seems to say that GOLD was predicting binding based on the experimental structure for a given ligand*

(for each ligand the authors removed the ligand from the experimental structure and ran GOLD to “predict” how the ligand binds). What were the results if GOLD were given just the InsP3(InsP4)/IP3KA structures with the ligands removed as the target?

*In the methods for docking, it says that docking was done in two ways:
1. Using the observed conformation of the enzyme corresponding to the particular ligand;
2. Using an ensemble of all the conformations observed for the enzyme.
However, the legend for Extended Fig. 4 seems to refer only to the first method. What were the results of the second method?*

Yes, on one hand, we have used the two mentioned approaches (1 and 2) for GOLD validation. On the other, predictions for novel protein-ligand complexes were made by the second approach (ensemble, results moved to Supplemental Fig. 7). An ensemble docking is one of the bests GOLD options when experimental protein-ligand structures are not available, in a sense that it uses a repertoire of known experimental structures therefore accounting for protein flexibility.

Certainly, we had included results of validation from approach 1 (Supplementary Figure 6a) whereas the results of the validation obtained from the approach 2 (ensemble) were missing. We have now included this data (Supplementary Fig. 6b), in which we can appreciate that this procedure yields a cluster of good solutions in all cases.

Following the reviewer comments, we have also performed docking for all ligands against protein structures with codes 1w2c (IP3K-AMPPNP/InsP₃) and 1w2d (IP3K - ADP/InsP₄). We used the same set up of the ensemble docking to make the results comparable. We have had quite good results in the case of 1w2c. We decided to include the results against 1w2c to show the improvement of using ensemble docking (lines 337-340, Supplementary Fig. 6b,c). Overall, the number of good hits increase using ensemble docking against all available structures (Supplementary Fig. 6b,c). Moreover, the ensemble docking is particularly good at finding a unique cluster of solutions as compared to docking against 1w2c (Supplementary Figure 6b), which find more clusters including worse solutions. In addition, we see that the ensemble docking predicts better the accurate InsP₃ site. We have added a conclusion in the main text (lines 337-340), as well as a detailed explanation in Supplementary Figure 6b,c.

15. Legend for the Ext Fig. 4b. The authors state:

“GOLD best solutions for compounds 7b-s, 8 and 9 bound in IP3K active site that show how phosphates mimic InsP3 phosphate positions. In these three cases the conformations of the ligands present non-favored torsions explaining why we have not detected significant binding in any of our experiments and they failed to form a crystal complex.”

Which torsions are the authors referring to? Are these the chair/boat/twist-boat conformations of the ligand?

Given the overall observation that GOLD was not able to dock product ligands confidently, even though it was given the conformation of the enzyme from the enzyme/product complex, it does not seem as though the docking pose predicted by GOLD for 7b-s, 8, 9, 10, and 11 should be taken too seriously.

It seems as though the authors regard failure to get a ligand/protein structure by crystallography as an indication that a compound is not a substrate. Fig. 6 shows a product peak for 7b-s in the HPLC analysis, suggesting that it is acting as a substrate. It is a reasonably common experience for even ligands that are good substrates of enzymes not to bind to an enzyme in crystals, due to crystallisation reagents or crystal packing.

In the docking, the ring torsions have been kept rigid, then we are not exploring conformations other than the chair form found in InsP₃ (included in Method line 838). We have examined the whole torsion deviation of the solutions with respect to dihedral angles observed in compounds from the CSD (Cambridge Structural Database), using MOGUL program (histograms of Supplementary Figure 7a). The results show that several dihedral angles of compounds 7b-a, 8 and 9 deviate significantly from normal values, making the poses obtained unlikely to be feasible solutions. We have properly commented upon this in the manuscript (lines 345-351).

We believe that compounds 7b-s, 8, 9, 10 and 11 could be substrate mimics, as they lack a phosphate in pseudo position 3. Some GOLD solutions successfully place the three phosphates of all five compounds in the correct places. However, for compounds 8 and 9 their molecular structures differ significantly from InsP₃, making docking predictions more speculative. This, together with the unusual torsions, persuade us not to consider the result as positive. In contrast, compounds 7b-s, 10 and 11 maintain sufficient similarity to the InsP₃ structure. We think that solutions for compounds 7b-s and 11 looks very reasonable and are comparable to the InsP₃ binding mode. However, as mentioned, 7b-s torsions are also outside normal values. We agree that compound 7b-s warrants a better discussion, since as the reviewer notes, 7b-s is a poor substrate. Certainly, the failure to produce an IP3K/7b-s crystal complex could be attributed to various factors. In here, it is important to note that crystal packing, reagents etc have not hindered crystal formation with other similar substrates. We have revised the text to provide a more comprehensive discussion for the results of 7b-s (page 344-348).

16. Line 472. The authors refer to “..., either primary or secondary position...” This would be better as “ ..., either as a primary or secondary hydroxyl, ...

Done (line 490)

Reviewer #2:

We are grateful to reviewer #2 for the nice comments about the work. We hope to answer satisfactorily the reviewer's concerns with the responses below:

My main concern is the lack of any comparisons with other IP-kinase such IPMK and ITPKs. To better appreciate the newly discovered IP3K enzymatic plasticity the introduction should compare IP3K substrate selectivity with the substrate 'promiscuity' of IPMK/ITPKs. Similarly, the discussion should comment (or speculate) on what the authors discovered for IP3K means for IPMK/ITPKs enzymology.

We agree with the reviewer that IPK family substrate selectivity should be commented in the introduction. We have added a paragraph (lines 68-75) about IPMK's promiscuity versus IP3K's high substrate selectivity. We also roughly mentioned the case of ITPKs without deepening in it due to its different fold.

We have also included a paragraph in the discussion (lines 405-408) suggesting that IPMK might present plasticity to recognize different ligands, but in a different region due to the IP-lobe differences. Structurally, IPMK does not conserve the plastic $\alpha 2I$, but conserve a similar but less constrained helix ($\alpha 4I$) that could accommodate different ligands. We compare the structures in Supplementary Figure 8 d. We also mentioned that other IPKs, as IPMK, might phosphorylate substrates with primary hydroxyls (line 463), a topic for future research.

We take this reviewer's concern fully on board and hope that in the near future we can pursue work with both suggested enzymes using our library of synthetic compounds for extra comparative clarity.

Line 57 please define CAM. It is +/- defined on line 61 where instead should appear as CAM.

We have corrected this mistake.

Lines 105-114 The authors should try to simplify the very long note in [bracket].

We have simplified the note in brackets to make it easier to follow (lines 111-118). We hope this text will help readers understand the isomerism and regiochemistry of these compounds compared to InsP₃.

Line 188 I would be more specific and define an equatorial 2OH or pseudo-2OH substituent.

Changed by "with either an equatorial OH or no substituent at this pseudo 2-position" (line 191)

Line 293 While the unexpected discovery that -InsP3 analogues substituted at the pseudo 3-position- is quite important, it does not need to be written in bold. It did not represent the title of a new paragraph.

Certainly. Bold letter has been removed now (line 297).

Figure 1 and extended Fig.1 I suggest boxing the bold -Natural substrate and Natural product- labels, to be consistent with the other boxed bold labels. Additionally, I would define also in the figure legend (not just as a figure's element) the meaning of the colored yellow, red, and green arrows. Only the green arrow is defined.

Thanks, we have corrected both things.

Reviewer #3:

If I were to request an addition/revision to the paper, I think some of the forward-looking statements about other substrates that could reveal new biochemistry could be less vague. What do these new results make the authors curious about? My guess is that they have lots of ideas, but maybe they could speculate more on what the specific results of this paper presage.

We thank the reviewer for the kind comments about the work.

We believe that, based on our results, novel ligands from these enzymes could be designed, which will require *in vivo* activity testing. Therefore, we foresee two potential lines of interest: the design of new ligands and the exploration of their cellular activities.

Regarding the design of new ligands, the most straightforward strategies would likely be to identify compounds able to target the InsP binding site of IP3K, exploring the conformational variability outlined in this work. Ensemble docking using virtual libraries could aid in selecting compounds that targets the active site. We find the prospect of obtaining selective ligands for different InsP binding enzymes particularly appealing, especially for crucial enzymes like IP3R, and also for other IPKs such as IPMK. In this regard, exploring InsP substituents at position 3 could yield specific inhibitors for IP3K, and not for IP3R, thus leading to selective inhibitors for InsP₃ binding enzymes. This approach will likely align with the design of bi-substrate inhibitors. As demonstrated in this work, we could also take the advantage of carbohydrate-based synthetic strategies to explore novel InsP analogues.

The ligands analyzed herein, and those to be designed, could have significant *in vivo* impacts, both for human health and cell biology studies. Often, proper regulation of InsP binding enzymes, including IPKs, is essential for cell functioning and disease prevention. For instance, in the last decade, IP3K has been associated with tumorigenesis and metastatic processes. Additionally, effective selective inhibitors would be valuable in delineating the pleiotropy of many of these enzymes, such as the IPK family, through cell biology studies.

We have checked that these ideas are indeed mentioned in the manuscript and, though only in speculative outline, we have added some lines at the end of the discussion stressing the importance of these molecules for designing potential therapies and for InsP binding enzymes *in vivo* cellular studies.

REVIEWERS' COMMENTS

Reviewer #1 (Remarks to the Author):

I thank the authors on carefully and fully addressing all my concerns in the revisions. I enthusiastically recommend the publication of this impressive work in Nature Communications.

Reviewer #2 (Remarks to the Author):

In the revised version of the manuscript, the authors have fully addressed my original request.